# Hierarchically structuralized hydrogels with ligament-like mechanical performance

Diwei Shi[1,4], Donghwan Ji [2,3,4] & Jinhye Bae [1,2,3] ✉

Mechanical properties of synthetic hydrogels remain inferior to those of load-bearing tissues such as ligaments, one of the strongest and stiffest natural hydrogels in the human body. Inspired by biological structures and their mechanisms conferring high mechanical properties, we report strong, stiff, and tough hydrogels composed of fiber-shaped elements that can be assembled into parallel bundles, closely resembling natural ligaments. These hydrogel fibers, readily fabricated with diameters of a few hundred micrometers, comprise polymer–particle hybrid agglomerates embedded in a continuous, interconnected polymer matrix. Strong polymer–particle interactions combined with spatial confinement within the agglomerates enable efficient load transfer, resulting in significant load-transfer lengths and substantial energy dissipation across the network. This design overcomes the conventional trade-offs between strength/stiffness and toughness/stretchability in polymer composites, thereby achieving tensile strength of $61 \pm 8$ MPa, elastic modulus of $131 \pm 15$ MPa, toughness $135 \pm 11$ MJ m$^{-3}$, and stretchability exceeding 400%. When assembled into millimeter-scale hierarchical bundles, the hydrogels mimic the structural organization of ligaments, sustain loads of tens of kilograms, and function as strain sensors.

In our bodies, strong, stiff, and tough ligaments connect bones to other bones, stabilizing joint movements and limiting motion within normal ranges (Fig. 1a)[1]. The ligaments comprise numerous individual collagen fibers, being dense connective tissues, which are capable of high mechanical loading[2,3]. These collagen fibers are organized in parallel bundles with a hierarchical aligned structure, thereby rendering the ligaments to possess enhanced mechanical performance compared to the individual fibers. Several ligaments, such as the anterior cruciate ligament, exhibit tensile strengths of ~50 MPa and elastic moduli of ~100 MPa[4,5]. Despite their inherent good strength and stiffness, ligaments are frequently injured due to trauma or excessive loading, especially when compromised by aging or abrupt movements[5,6]. Treating ligament injuries, such as reconstructing their dense, hierarchically aligned structures to restore their original mechanical functions, is a prolonged process[7]. To facilitate this recovery, bridging ligament defects or replacing the ligament using a graft material is often clinically implemented[8]. However, mismatches in mechanical properties between graft materials and natural ligaments remain a limitation[9].

Recent efforts have focused on developing synthetic hydrogels that mimic the mechanical properties of biological tissues (i.e., natural hydrogels) like ligaments[10,11]. Synthetic hydrogels, comprising polymeric networks and water, are recognized for their biocompatibility and potential to support the repair of injured biological tissues[12,13]. Particularly, matching mechanical properties between synthetic hydrogels and biological tissues is a crucial consideration, and strong, stiff, and tough ones could supplement substantial mechanical loading subjecting to the injured ligaments[14]. In this context, various strategies have been explored to achieve enhanced mechanical properties in synthetic hydrogels, including the formation of crystalline domains[15,16], supramolecular networks[17], interpenetrating networks[18], topological entanglements[19], or nano-/micro-fibrous structures[20,21], or the

[1]Materials Science and Engineering Program, University of California San Diego, La Jolla, CA, USA. [2]Aiiso Yufeng Li Family Department of Chemical and Nano Engineering, University of California San Diego, La Jolla, CA, USA. [3]Department of Chemical Engineering, Chung-Ang University, 84 Heukseok-ro, Dongjak-gu, Seoul, Republic of Korea. [4]These authors contributed equally: Diwei Shi, Donghwan Ji. ✉e-mail: jbae@cau.ac.kr

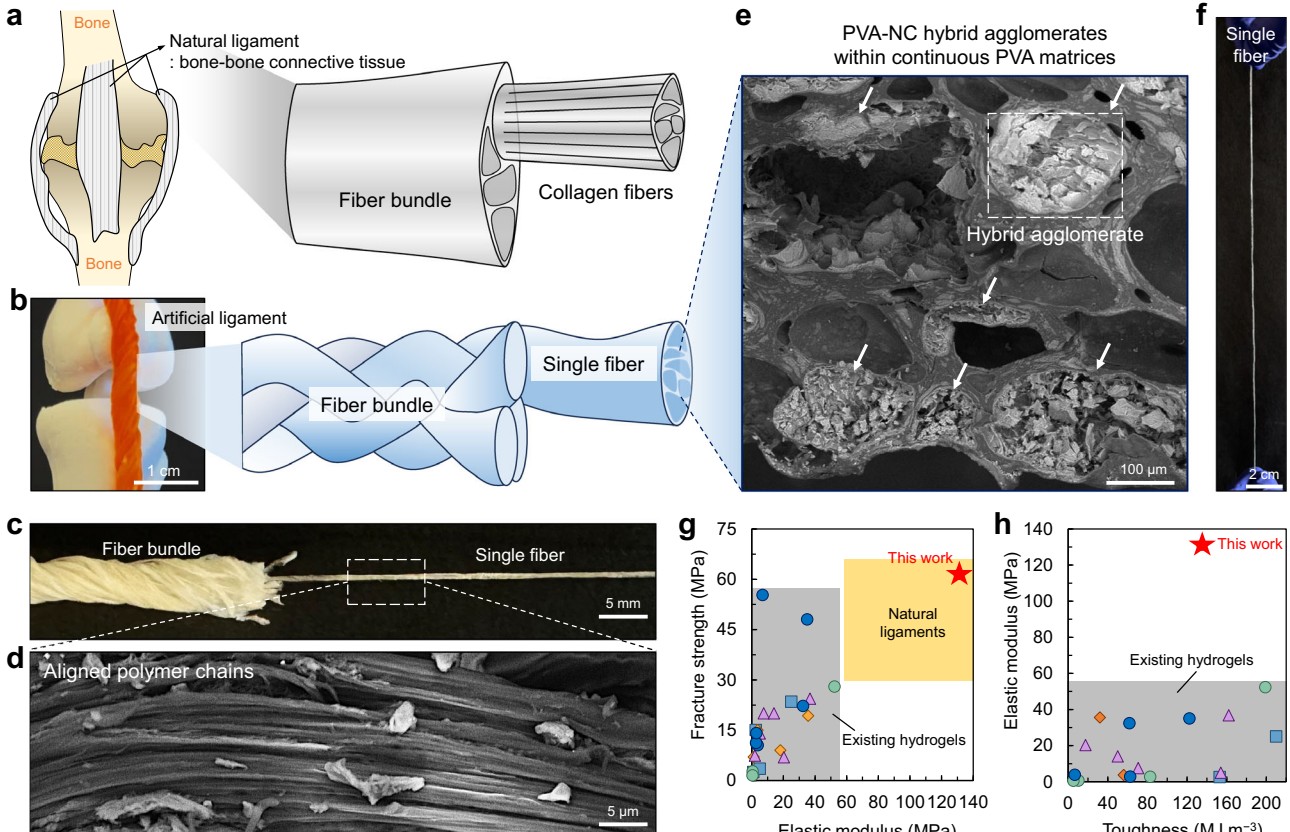

**Fig. 1 | Ligament-mimicking hierarchically structuralized hydrogels.** Schematic illustrations of (**a**) natural ligament and (**b**) artificial ligament, displaying their hierarchical aligned structure comprising fibers. **c** Photograph of a hydrogel bundle as an artificial ligament comprising multiple braided fibers. Scale bar, 5 mm. **d** Cross-sectional scanning electron microscopy (SEM) image of the single hydrogel fiber, parallel to the extrusion direction, demonstrating aligned polymer chains. Scale bar, 5 μm. **e** Cross-sectional SEM image of the single hydrogel fiber, perpendicular to the extrusion direction, demonstrating PVA–NC hybrid agglomerates within continuous PVA matrices. The arrows indicate the hybrid agglomerates. Scale bar, 100 μm. **f** Photograph of a single hydrogel fiber. Scale bar, 2 cm. **g, h** Diagrams showing tensile strength, elastic modulus, and toughness of PVA–NC/FT/S hydrogel (this work), natural ligaments, and previously reported mechanically reinforced PVA-based hydrogels. All the detailed mechanical properties and their literature are summarized in Supplementary Table 1.

incorporation of nano-/micro-particles into the hydrogel matrix[22]. Notably, recent studies have demonstrated that combining freeze-casting with the Hofmeister effect (i.e., salting-out effect) on polyvinyl alcohol (PVA) hydrogels can result in high mechanical properties[15]. In addition, subsequent studies have reported strong and tough PVA hydrogels by employing complementary techniques, such as the aforementioned methods, solvent exchange, freezing-thawing, and flow-induced alignment[15,23–27]. Nevertheless, the mechanical properties of these synthetic hydrogels, particularly strength and stiffness, remain lower than those of natural ligaments; therefore, achieving ligament-like mechanical performance in synthetic hydrogels has been a challenge.

In this study, we present ligament-mimetic strong, stiff, and tough hydrogels comprising hydrogel fibers organizable in parallel bundles with a hierarchical aligned structure (Fig. 1b–d). The ligament-level mechanical performance is achieved by integrating polymer-nanoparticle hybrid agglomerates within a continuous polymer matrix (Fig. 1e). PVA and nanoclay (NC) serve as the primary components for fabricating hydrogel fibers (Fig. 1f). NC acts as both a mechanical reinforcement and a rheological modifier, rendering the initial aqueous PVA–NC mixture a viscoelastic solid that is extrudable into mechanically stable fibers. During extrusion, the PVA and NC components, initially mixed randomly, undergo shear-induced alignment. The extruded PVA–NC mixture fibers are then subjected to freezing, thawing, and treatment with an aqueous salt solution,

resulting in hydrogel fibers with diameters of 450–500 μm. During the freezing-thawing process, NC particles agglomerate within the PVA matrix, forming a distinct structure with PVA–NC hybrid agglomerates incorporated within the continuous PVA matrix (Fig. 1e). Strong interfacial interactions between PVA and NC and their spatial confinements within the agglomerates simultaneously enhance strength, stiffness, and toughness without compromising stretchability. Consequently, the PVA–NC hydrogel fiber demonstrates significantly enhanced mechanical properties: a tensile strength of $61 \pm 8$ MPa, an elastic modulus of $131 \pm 15$ MPa, and a toughness of $135 \pm 11$ MJ m$^{-3}$ (Fig. 1g, h, Supplementary Table 1). Furthermore, these PVA–NC hydrogel fibers can be constructed into parallel bundles with a hierarchical aligned structure that emulates ligament organization. These braided fibers exhibit great mechanical performance compared to individually separated fibers, and a fiber-braided bundle with ~3.5 mm diameter is capable of lifting at least 13.6 kg weight, highlighting its high mechanical performance.

## Results and discussion

### Design and fabrication of PVA–NC hydrogel fibers

The high mechanical performance of ligaments arises from the collagen fibers organized into parallel bundles with a hierarchical aligned structure[3,28]. Additionally, partially mineralized collagen-based tissues (e.g., enthesis) exhibit greater load-bearing capacity and higher stiffness than unmineralized ones, in which stiff mineralized collagens (i.e.,

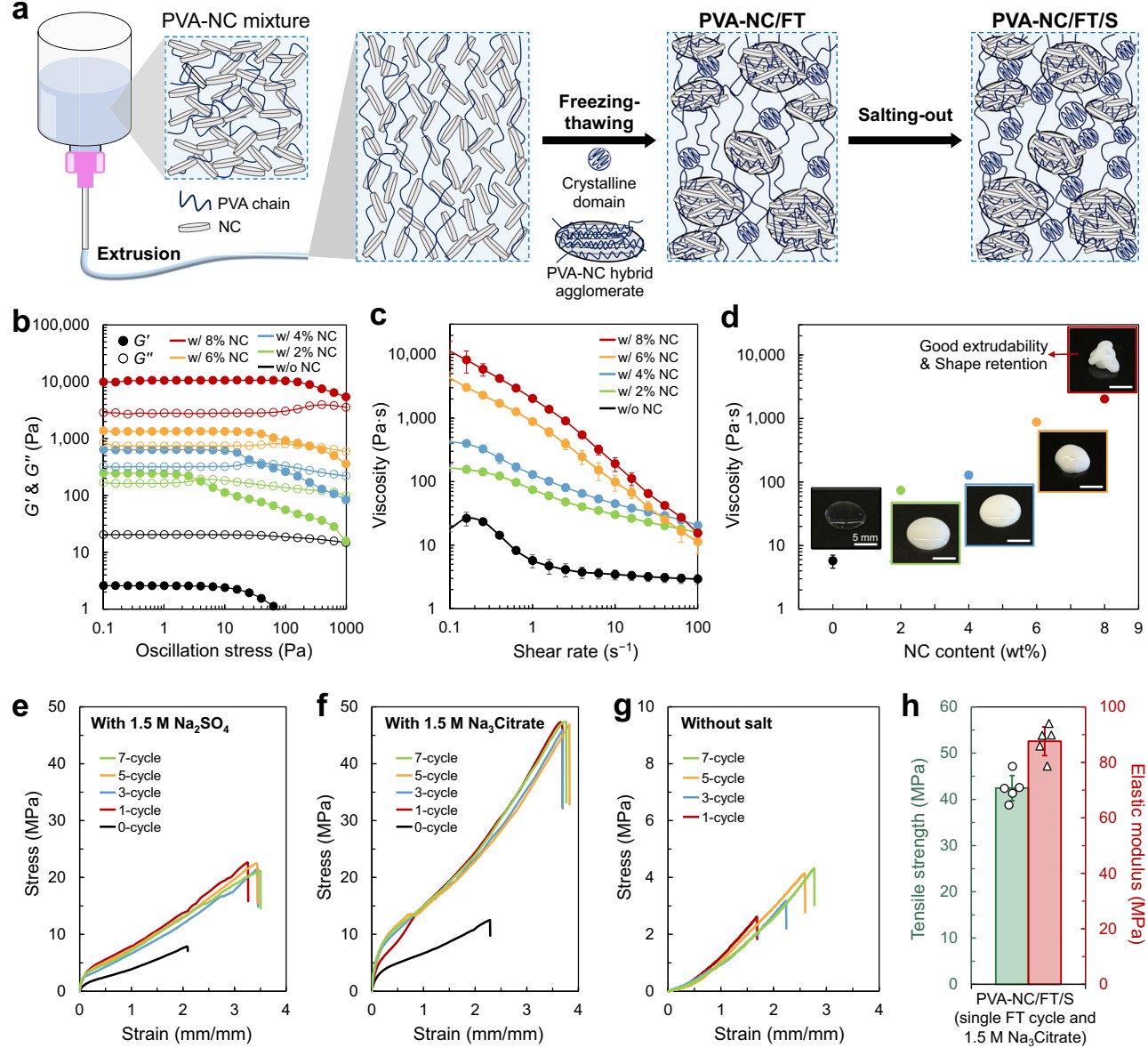

**Fig. 2 | Preparation and mechanical properties of hydrogel fibers. a** Schematic illustration depicting hydrogel fiber fabrication procedure: extrusion, freezing-thawing (FT), and salting-out (S) processes. **b** Storage modulus ($G'$) and loss modulus ($G''$) as a function of oscillation stress and (**c**) viscosity as a function of shear rate for PVA–NC mixtures with different NC content. **d** Viscosity and photographs of PVA–NC mixtures with different NC contents. The viscosity value is a representative value at the shear rate of $1\,s^{-1}$. Scale bar, 5 mm. Stress-strain curves of PVA–NC/FT/S hydrogel fibers formed with varying FT cycles and S treatment by a different salt solution type: (**e**) 1.5 M $Na_2SO_4$, and (**f**) 1.5 M $Na_3$Citrate. **g** Stress-strain curves of PVA–NC/FT hydrogel fibers formed with varying FT cycles without salting-out. **h** Tensile strength and elastic modulus of the PVA–NC/FT/S hydrogel fibers processed by a single FT cycle and 1.5 M $Na_3$Citrate salt solution. Data are presented as mean ± s.d. ($n = 3$).

mineral-collagen complexes) are intertwined within flexible collagen matrices[29–31]. Inspired by these biological structures and mechanisms, which confer high mechanical properties, we designed ligament-like strong, stiff, and tough hydrogels comprising hydrogel fibers enhanced by a distinct structure of polymer-nanoparticle hybrid agglomerates within continuous polymer matrices.

To fabricate strong, stiff, and tough hydrogel fibers, we employed a combination of PVA and NC with freezing-thawing (FT) and salting-out (S) processes (Fig. 2a). In conventional methods, liquid-like PVA solutions should be directly injected into a coagulation bath (e.g., high-concentration salt solution) to form PVA networks solely via the S process without FT process, resulting in solidified hydrogel fibers[21,22,24]. By contrast, the viscoelastic PVA–NC mixture was extrudable into long fiber shapes without a coagulation bath because NC served as both a

mechanical reinforcement and rheological modifier, imparting a viscoelastic solid-state property[32]. During extrusion, the PVA and NC components were subjected to shear-induced alignment (Fig. 1d)[21,33]. Following extrusion, the PVA–NC fibers were treated with sequential FT and S processes, resulting in PVA–NC/FT/S hydrogel fibers with 450–500 μm diameters using a 600 μm nozzle. During the FT process, NC particles agglomerated, forming hydrogels with PVA–NC hybrid agglomerates and continuous polymer matrices (Fig. 1e). We expected that the combination of sequential FT and S treatments forming a distinct structure, composed of PVA–NC hybrid agglomerates within the continuous polymer matrix with aligned polymer chains, would enhance the mechanical properties of hydrogels[29,34–38].

To confirm the effect of NC on PVA solution and access the potential for fiber formation, we first investigated the rheological

properties of PVA–NC mixtures with varying NC content (2–8 wt%). Unlike pure PVA solution, which exhibits a greater loss modulus ($G''$) than storage modulus ($G'$), the PVA–NC mixtures demonstrated a viscoelastic solid behavior with $G'$ exceeding $G''$ in the absence of external force (Fig. 2b). When subjected to external force excessing a critical level, the mixtures underwent shear-yielding, suggesting their good extrudability[33,39]. These mixtures also exhibited the shear-thinning behavior (Fig. 2c). While the PVA–NC mixture has high viscosity at low shear rates (i.e., before and after extrusion), the mixture temporarily has low viscosity under high shear rates during extrusion. Among the tested formulations, the PVA–NC mixture with 8 wt% NC demonstrated the most appropriate viscosity and viscoelastic properties, facilitating smooth extrusion through syringe nozzles and retaining its shape after extruded (Fig. 2d). Mixtures with less than 8 wt% NC lost their shape in 5 min after extruded (Fig. 2d, inset photographs), while mixtures with more than 8 wt% NC were too thick to form long fibers through continuous extrusion.

Using the PVA–NC mixture with 8 wt% NC content, we fabricated PVA–NC hydrogel fibers treated with various FT cycles and different salt types to evaluate the effect of FT and S processes on mechanical properties. Sodium sulfate ($Na_2SO_4$) and sodium citrate ($Na_3Citrate$), known for their strong salting-out effect[15], were used at a fixed concentration of 1.5 M in this evaluation. PVA–NC/S hydrogel fibers, treated only with salting-out directly without FT cycles (i.e., 0 cycle), were significantly weaker and softer than the PVA–NC/FT/S hydrogel fibers (Fig. 2e, f, black lines). This finding demonstrates that salting-out alone has a limitation in achieving high mechanical properties. Overall mechanical properties (fracture strength and strain, and elastic modulus) of PVA–NC/FT/S hydrogel fibers remained generally consistent across one to seven FT cycles. However, they were significantly affected by the salt type due to variations in salting-out efficiencies[15] (Fig. 2e, f). Specifically, $Na_3Citrate$-treated hydrogel fibers exhibited significantly higher fracture strength and elastic modulus compared to $Na_2SO_4$-treated hydrogel fibers (Supplementary Fig. 1); thus, $Na_3Citrate$ was the most effective salt in conferring high mechanical properties.

Conversely, PVA–NC/FT hydrogel fibers fabricated without salting-out exhibited lower mechanical strength, stiffness, and stretchability (Fig. 2g). While increasing the number of FT cycles slightly improved stretchability, the mechanical properties of PVA–NC/FT fibers were an order of magnitude inferior to those of the PVA–NC/FT/S fibers. These results underscore the importance of the sequential combination of FT and S processes in achieving enhanced mechanical properties. Notably, even a single FT cycle followed by $Na_3Citrate$ treatment is both efficient and effective in producing mechanically robust hydrogel fibers with a tensile strength of 42.4 MPa and an elastic modulus of 87.6 MPa (Fig. 2h). Because pure PVA solution and PVA–NC mixtures with a low NC content could not be used for the fiber fabrication, subsequent experiments focused on evaluating the mechanical properties and mechanisms of hydrogel fibers, PVA–NC/FT/S at the fixed 8 wt% NC, treated with varying $Na_3Citrate$ concentrations after one FT cycle.

## Enhanced mechanical properties and reinforcing mechanisms

We evaluated the effect of $Na_3Citrate$ concentration (0.1–2.8 M) on the mechanical properties and achieved a combination of high strength, stiffness, and toughness, comparable to natural ligaments (Fig. 3a–c). As the $Na_3Citrate$ concentration increased, the mechanical properties of the PVA–NC/FT/S hydrogel fibers improved proportionally. At the saturated $Na_3Citrate$ concentration of 2.8 M (at 20–25 °C), the hydrogel fiber exhibited a fracture strength of 61 ± 8 MPa, an elastic modulus of 131 ± 15 MPa, and a toughness of 135 ± 11 MJ m$^{-3}$. A single PVA–NC/FT/S hydrogel fiber with ~450 µm in diameter was capable of lifting a ~400 g weight without deformation (Supplementary Movie 1), demonstrating better mechanical robustness over PVA–NC/FT and

PVA/S hydrogel fibers (Supplementary Movie 2). The mechanical properties of PVA-based hydrogels are summarized in Fig. 1g, h, Supplementary Table 1 (ref[4,15,21,22,24,40–63]). Particularly, unlike previous hydrogels with elastic moduli in the one-to-ten MPa range, our hydrogels achieved elastic modulus in the hundred MPa with commendable toughness over the hundred MJ m$^{-3}$, aligning with natural ligament values[4,5]. In addition, the mechanical properties of PVA–NC/FT/S hydrogel fibers with the aligned polymer structure were higher than those of a hydrogel sheet formed by casting (Fig. 3d, Supplementary Fig. 2). Such greater stiffness and toughness of the fiber than those of the sheet elucidated the role of shear-induced alignment[15,16,21].

X-ray diffraction (XRD) analysis confirmed the formation of well-defined crystalline domains within the PVA networks, indicating a continuous and robust polymer matrix established in the presence of NC (Fig. 3e). The characteristic PVA crystalline peak at $2\theta = 19.8°$ was evident in PVA–NC/FT/S, and the PVA crystalline peak remained significant and dominant over NC peaks, indicating that NC did not hinder the formation of PVA crystalline domains. Further, the presence of larger crystalline domains in PVA–NC/FT/S, evidenced by a sharper PVA crystalline peak compared to that of PVA–NC/FT (Fig. 3f), supports higher mechanical properties of PVA–NC/FT/S than those of PVA–NC/FT (Fig. 2f, g)[15,16,41]. This observation also correlates with the difference in water content between PVA–NC/FT/S and PVA–NC/FT (Supplementary Fig. 3). The larger crystalline domains indicate closer packing and stronger fastening of polymer chains within the hydrogel, which explains the lower water content observed in PVA–NC/FT/S than in PVA–NC/FT.

Moreover, Fourier transform infrared (FTIR) spectroscopy analysis demonstrated the interactions between PVA and NC (Fig. 3g). Compared to PVA/FT/S, PVA–NC/FT/S exhibited an amplified peak intensity at 3400 cm$^{-1}$, indicating increased O–H stretching vibrations[16,64]. While NC displayed two prominent peaks at ~3600 and 3400 cm$^{-1}$, the 3600 cm$^{-1}$ peak was absent in the PVA–NC/FT/S hydrogel, suggesting that the 3400 cm$^{-1}$ peak predominantly arises from interactions between PVA and NC. This interaction likely results from the agglomeration of PVA and NC, which expels water molecules previously bound to hydrated PVA chains, thereby revealing hidden O–H stretching vibrations. Additionally, the slight increase in peak intensity at a lower wavenumber (~3170 cm$^{-1}$) implies weakened O–H stretching due to the formation of hydrogen bonding between PVA and NC[65,66]. Differential scanning calorimetry (DSC) analysis further confirmed substantial binding interactions between PVA and NC (Fig. 3h). As the NC content increased from 0 to 8 wt%, the PVA melting temperature gradually shifted upward, accompanied by a reduction in melting peak sharpness and intensity. At 8 wt% NC, the melting peak was significantly broadened and not precisely defined, reflecting significantly restricted thermal motion of PVA chains. This reduction in chain mobility was attributed to the spatial confinement of PVA chains between NC, a phenomenon intensified in the PVA–NC agglomerates (Fig. 1e)[67–69].

Based on these results, including FTIR, DSC, and SEM analyses, the mechanical reinforcement and fracture mechanisms of the PVA–NC/FT/S hydrogels are proposed as follows (Fig. 3i). Upon tensile loading, the applied stress is effectively transferred from the flexible PVA matrix to the rigid PVA–NC agglomerates through strong cohesive interactions, including hydrogen bonding and spatial confinement between PVA chains and NC particles[36–38,70]. Because the PVA–NC hybrid agglomerates are intertwined within the continuous PVA matrix and the PVA chains are spatially connected across the agglomerates and matrices, stretching of PVA chains under the loading accompanies the deformation of agglomerates. Over these extended load-transfer lengths, substantial energy dissipation occurs[30,31,36,37], resulting in the hydrogel's high fracture strength and strain (Fig. 3a). With further stretching, microcracks are likely to initiate within the PVA matrix and propagate along the agglomerate surfaces, delaying ultimate

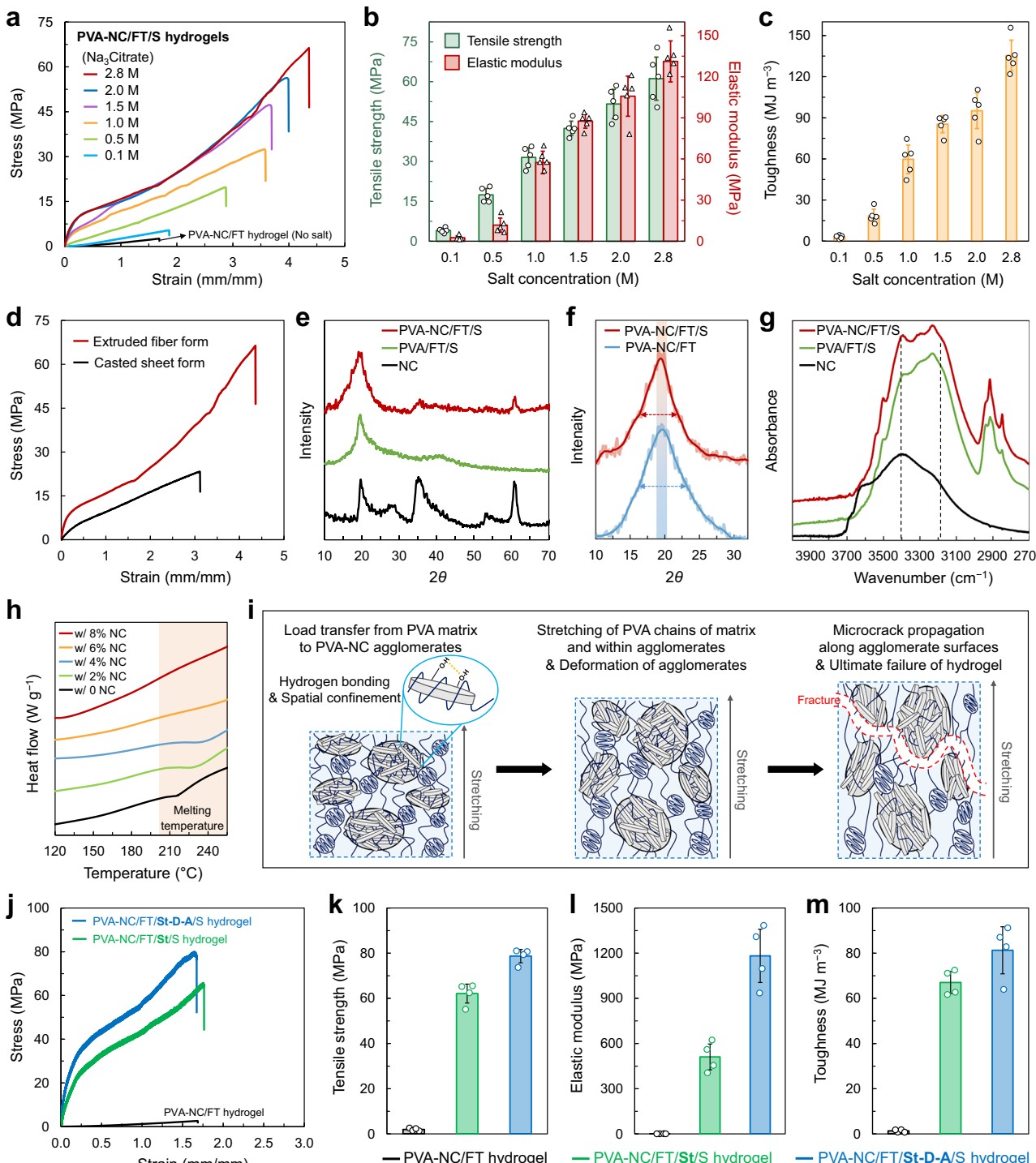

**Fig. 3 | Enhanced mechanical properties and reinforcing mechanisms. a** Stress-strain curve, (**b**) tensile strengths and elastic moduli, and (**c**) toughness of PVA−NC/FT and PVA−NC/FT/S hydrogel fibers at varying concentrations of Na₃Citrate. Data are presented as mean ± s.d. (*n* = 5). **d** Stress-strain curve of PVA−NC/FT/S hydrogels in extruded fiber form and casted sheet form, respectively. **e** XRD spectra of NC, PVA/FT/S, and PVA−NC/FT/S, demonstrating the formation of PVA crystalline domains in the presence of NC. **f** PVA crystalline peak from XRD spectra of PVA−NC/FT and PVA−NC/FT/S. Arrows indicate the full width at half maximum. **g** FTIR spectra of NC, PVA/FT/S, and PVA−NC/FT/S, demonstrating the hydrogen bonding between PVA and NC. **h** DSC spectra of PVA−NC/FT/S with different NC contents, revealing melting temperature differed by the NC content. **i** Schematic illustration of mechanical reinforcement and fracture mechanisms of the PVA−NC/FT/S hydrogel. **j** Stress-strain curves, (**k**) tensile strengths, (**l**) elastic moduli, and (**m**) toughness of PVA−NC/FT, PVA−NC/FT/St/S and PVA−NC/FT/St-D-A/S hydrogel fibers. Data are presented as mean ± s.d. (*n* = 4).

failure[36,37]. This sequential fracture mechanism imparts significant tensile strength and elastic modulus while impeding failure and enhancing toughness, overcoming the conventional trade-off between strength/stiffness and toughness/stretchability.

Importantly, these reinforcement mechanisms differ from those of conventional inorganic platelet-reinforced composites. In typical composites, inorganic platelets enhance strength and stiffness primarily through load transfer described by the shear-lag model, where

applied stress is predominantly transferred from the polymer matrix to individual reinforcements via interfacial shear[67,71–73]. Under this mechanism, the energy dissipation zone (or process/bridging/damage zone) is inherently limited, resulting in restricted toughness and stretchability[67,74–76]. By contrast, the PVA−NC/FT/S hydrogels feature polymer−particle hybrid agglomerates embedded within a continuous polymer matrix, where polymer chains are spatially interconnected across the agglomerates and matrices throughout the entire hydrogel. This distinct structure enables effective load transfer across long distances, which leads to high toughness and stretchability. If NC were only individually dispersed or poorly integrated in the matrix, load transfer would be confined to short distances, substantially diminishing mechanical reinforcement, in particular, toughness. This structural design resembles that of partially mineralized tissues–an intermediate between unmineralized tissue and fully mineralized tissue–which are known to achieve high strength, stiffness, and toughness beyond predictions of classical theories like the shear-lag model[77–79]. Therefore, the distinct internal structure enabled our hydrogel to simultaneously achieve high strength, stiffness, toughness, and stretchability, breaking the traditional trade-off observed in polymer composites.

Furthermore, we achieved additional improvements in the mechanical performance of the hydrogel, particularly in strength and stiffness (Fig. 3j–m), by integrating previously established stretching and dry-annealing processes[41,80–82]. Specifically, the PVA−NC/FT hydrogel fiber was first stretched to 100% deformation; the fracture elongation of PVA−NC/FT was approximately 150–160% (Fig. 2g). This stretched sample (PVA−NC/FT/St) was then immersed in 2.8 M $Na_3$Citrate solution for the salting-out process, yielding a PVA−NC/FT/St/S hydrogel fiber. For a PVA−NC/FT/St-D-A/S hydrogel fiber, the stretched sample was additionally dried and annealed at 70 °C for one day before the salting-out process. As a result of polymer chain alignment and densification induced by the stretching and dry-annealing processes[41,64,73,80,82], both PVA−NC/FT/St/S and PVA−NC/FT/St-D-A/S hydrogel fibers exhibited an elastic modulus of 511 MPa and 1182 MPa, respectively, while still maintaining good stretchability beyond 150% deformation (Fig. 3j–m). Notably, for the PVA−NC/FT/St-D-A/S hydrogel, fracture strength, elastic modulus, and toughness are 40.2-, 1225-, and 61.2-fold higher, respectively, than those of the PVA−NC/FT hydrogel.

### Hierarchically structuralized hydrogel fiber bundles mimicking ligaments

Mimicking the natural ligaments, composed of collagen fibers organized into parallel bundles, we fabricated hydrogel bundles by braiding multiple hydrogel fibers (Fig. 1b). Owing to the simplicity of the fabrication process (no special instruments, harmful chemicals, etc.), scalability, and robustness of individual fibers, we readily produced hydrogel bundles in varying sizes. Specifically, PVA−NC/FT fibers (e.g., from 2 to 8 fibers) were braided into a bundle and submerged in 2.8 M $Na_3$Citrate solution for the salting-out process, which facilitated close contact between each fiber and formed uniform hydrogel bundles (Fig. 4a). The eight-fibers braided bundle exhibited two-fold higher mechanical loading capacity compared to the individually separated 8-fibers, respectively (Fig. 4b). The separated fibers were subjected to uneven stress under stretching, thereby resulting in early fracture and lower mechanical loading capacity than the braided fibers (Fig. 4c). In the braided case, stress was effectively distributed across multiple fibers in the bundle[80,83], resulting in neat fracture of the whole bundle (Fig. 4d). The structural integration of braided fibers enables the load sharing, similar to previous findings in braided/coiled structures[80]. Using this strategy, larger bundles, such as a 50-fiber-braided bundle, were fabricated to closely emulate a natural ligament structure (Fig. 4e). This highly structured bundle exhibited good mechanical performance, withstanding a 30-pound dumbbell (i.e., a load of 13.6 kg) (Supplementary Movie 3).

In addition to their robust mechanical properties, the hydrogel bundles demonstrated the potential for a strain sensor (Fig. 4f−i, Supplementary Fig. 4). For instance, when an 8-fiber-braided hydrogel bundle was applied to a human skeleton model, including the leg (Fig. 4f) and wrist (Fig. 4h) as an artificial ligament connecting bone-to-bone, it detected resistance changes ($\Delta R$) during repeated flexion−extension (i.e., stretching−releasing) motions, enabling real-time monitoring of deformations. Leg movements requiring larger deformations resulted in greater $\Delta R$ values (Fig. 4g), while small deformations at the wrist resulted in smaller $\Delta R$ values (Fig. 4i). Notably, the high stiffness of the bundles allowed for precise detection of deformations due to small and stepwise flexion−extension motions, as demonstrated by the consistent resistance changes observed during small wrist movements (Fig. 4i). Specifically, small deformations at 35° produced minimal $\Delta R$, while stepwise deformations formed from 35° to 65° led to proportional, stepwise $\Delta R$ increases. This strain-sensing capability highlights the multifunctionality of the hydrogel bundles, making them promising candidates for biomedical applications with the ability to precisely monitor strain in real time, as well as ligament treatments. Their potential use in wearable sensing applications offers a dual function of mechanical support and deformation detection.

## Discussion

We have developed ligament-mimetic hydrogels with high strength, stiffness, and toughness, achieved through hydrogel fibers organized in parallel bundles with a hierarchically aligned structure. The fabrication of hydrogel fibers involves extrusion, freezing-thawing and salting-out processes. The shear-induced extrusion aligned polymer chains, and the subsequent FT and S processes induced the formation of PVA−NC hybrid agglomerates within a continuous PVA matrix. During these processes, NC did not simply separate from the matrix but instead aggregated within the PVA matrix, forming PVA−NC hybrid agglomerates incorporated in the continuous, interconnected PVA matrix. The applied load was effectively transferred to these agglomerates through strong cohesive interactions and spatial confinement between PVA chains and NC particles, enabling substantial energy dissipation over extended load-transfer lengths. Therefore, the PVA−NC/FT/S hydrogel fibers demonstrated tensile strength (61 ± 8 MPa), elastic modulus (131 ± 15 MPa), and toughness (135 ± 11 MJ m$^{-3}$), resembling natural ligaments. In addition, by integrating stretching and dry-annealing processes to enhance polymer chain alignment and densification, the hydrogel fibers achieved elastic modulus in the range of several hundred to over one thousand MPa, while still maintaining good stretchability beyond 150% deformation. Furthermore, fiber-braided hydrogel bundles with ligament-like organization exhibited higher mechanical performance than the individually separated fibers. The hydrogel bundles, consisting of a tunable number of fibers, would have a desirable mechanical loading capacity similar to a ligament, performing differently at different parts of our bodies. In addition to their enhanced mechanical performances, the hydrogel bundles were useful for precise detection of movement and deformation with resistance change and demonstrated the potential for strain sensors.

The good extrudability of the PVA−NC mixture was also applicable for fabricating a complex three-dimensional (3D) structure via a 3D printer (Supplementary Movie 4). The as-prepared PVA−NC mixture was printable in diverse shapes beyond the fiber form. This versatility enables the creation of hydrogels in customized shapes (Supplementary Figs. 5, 6), making the material adaptable to a wide range of applications. For enhanced practical usability, the hydrogels can be coated or encapsulated with soft, thin elastomer layers[84]. Because the salting-out effect primarily induces physical crosslinking within the PVA network, exposure to pure water leads to swelling and partial de-crosslinking, thereby weakening the mechanical properties (Supplementary Fig. 7). However, these properties can be effectively

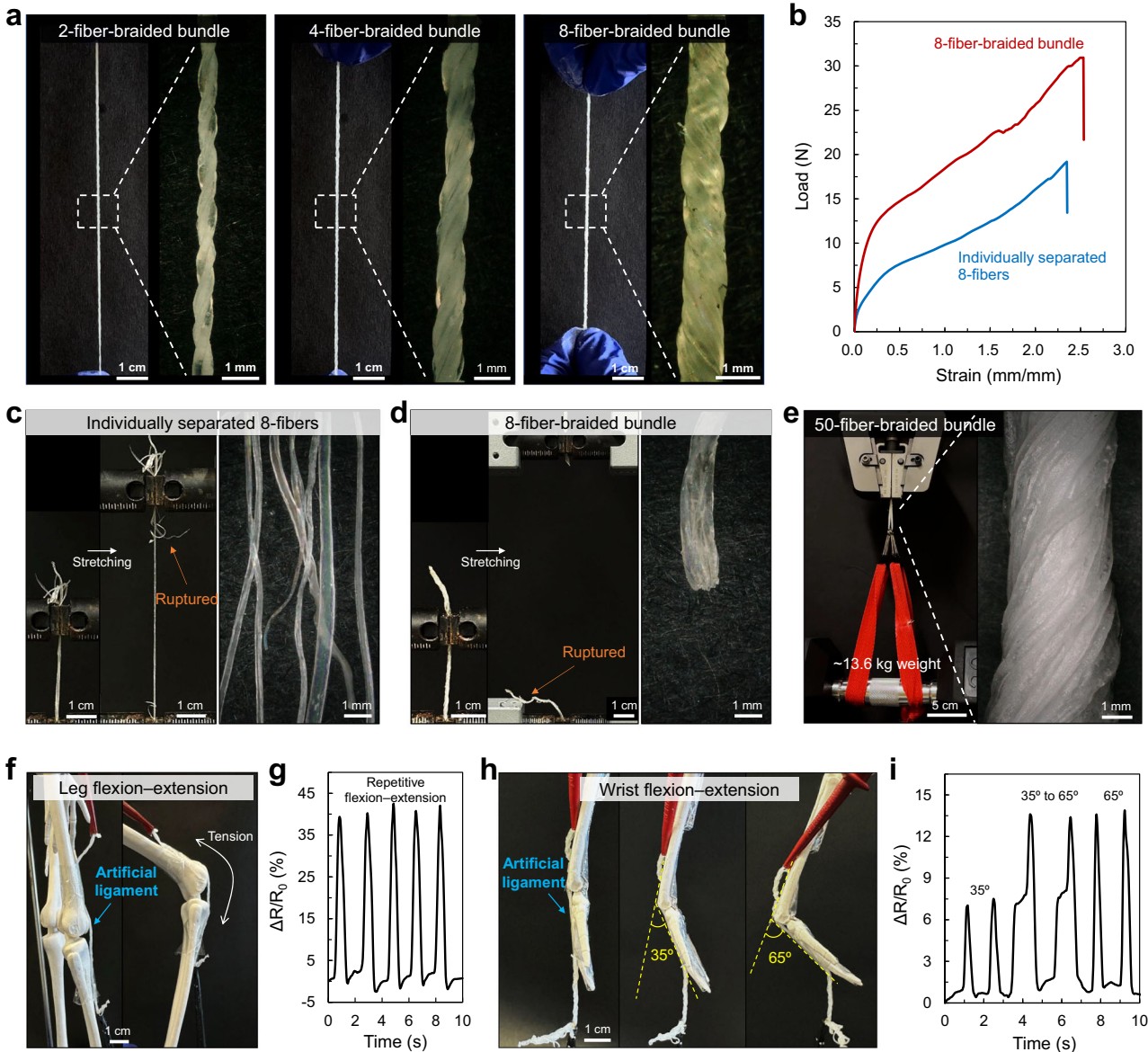

**Fig. 4 | Hierarchically structuralized hydrogel bundles mimicking ligaments.**
**a** Photographs of hydrogel bundles formed with multiple (two to eight) fibers. Scale bars from left to right, 1 cm and 1 mm, respectively. **b** Load-strain curves of single fiber, individually separated 8-fibers, and 8-fiber-braided bundle. Photographs of (**c**) individually separated 8-fibers and (**d**) 8-fiber-braided bundle, respectively, presenting a different fracture behavior under tension. Scale bars from left to right, 1 cm, 1 cm, and 1 mm, respectively. **e** Photograph of 50-fiber-braided hydrogel bundle withstanding ~13.6 kg weight. Scale bars from left to right, 5 cm and 1 mm, respectively. **f** Photographs and (**g**) real-time resistance changes of the 8-fiber-braided hydrogel bundle as an artificial ligament of the leg under repetitive folding and unfolding. Scale bar, 1 cm. **h** Photographs and (**i**) real-time resistance changes of the hydrogel bundle as an artificial ligament of the wrist at different bending angles. Scale bar, 1 cm.

preserved when the hydrogel is protected by a hydrophobic layer (e.g., elastomer film and oil layer) (Supplementary Fig. 7). Given their ligament-level mechanical performance, these hydrogels hold significant potential for diverse applications, including soft robotics, conductive gel electrolytes, and bioengineering uses.

## Methods
### Materials
Poly(vinyl alcohol) (PVA, $M_w = 89,000-98,000$, 99 + % hydrolyzed), sodium sulfate ($Na_2SO_4$, assay ≥ 99.0%), hydrochloric acid (37 wt% in water) and glutaraldehyde solution (25% in water) were purchased from Sigma Aldrich. Sodium citrate dihydrate ($Na_3Citrate$, assay ≥ 99.0%) and mineral oil were purchased from Fisher Scientific. Nanoclay (NC, Laponite-RD) was purchased from BYK BYK-Chemie. As-received dry NC powder was a large lump (Supplementary Fig. 8). All chemicals were used as received without further purification. Water is Milli-Q water.

### Preparation of PVA−NC and pure PVA hydrogels
An aqueous 15 wt% PVA solution was first prepared by mixing 15 g PVA with 85 g water at 100 °C for 3 h. For PVA−NC mixtures with a 2, 4, 6, 8, or 10 wt% NC content, 0.204, 0.416, 0.625, 0.869, or 1.111 g NC was thoroughly mixed with the 10 g of 15 wt% PVA solution using a thinky mixer. The resultant mixture was loaded into a 10 mL plastic syringe, and bubbles in the syringe were removed using the centrifuge at ~2500 × g for 10 min. The PVA−NC mixture was extruded into a fiber shape onto a flat glass plate using a 600 μm-diameter nozzle (Fig. 2a, extrusion process) unless otherwise defined. The extruded PVA−NC mixture underwent freezing (8 h)−thawing (8 h) for 1−5 cycles (Fig. 2a, freezing-thawing process for PVA−NC/FT hydrogels), and

subsequently was immersed in a salt solution for the salting-out treatment for over 24 h (Fig. 2a, salting-out process for PVA−NC/FT/S hydrogels). For the PVA−NC/FT/St/S hydrogels, the PVA−NC/FT hydrogel was first stretched to a 100% deformation (i.e., stretch process), and the ends of the stretched hydrogel were fixed to maintain the extended length. The sample was then immersed in a salt solution for over 24 h as the salting-out treatment. In the case of PVA−NC/FT/St-D-A/S hydrogels, the PVA−NC/FT hydrogel was first stretched to a 100% deformation (i.e., stretch process), the ends of the stretched hydrogel were fixed, and then the stretched hydrogel was dried in an oven at 70 °C for a day (i.e., dry-annealing process); subsequently, the dried sample was immersed in a salt solution for the salting-out treatment for over 24 h (i.e., salting-out treatment). In the case of pure PVA hydrogels, the as-prepared PVA solution was cast on a flat glass substrate in a sheet form and then underwent freeze-thawing and salting-out treatments because the pure PVA solution had low viscosity and was not free-standing in a fiber shape, unlike the PVA−NC mixture.

### Rheological and mechanical characteristics

Rheological characteristics, viscosity changes over the shear rate (flow sweep), moduli changes over the frequency change (frequency sweep), and moduli changes over oscillation shear stress (stress sweep), were performed using a rheometer with a 40 mm cone plate and a 52 µm truncation gap (TA Instrument, Discovery HR-3). The PVA and PVA−NC composite samples were placed on a 20 °C Peltier flat stage and temperature-equilibrated for 60 s.

Tensile mechanical tests were performed using an Instron 5982 universal testing machine with a 100 N load cell at 50 mm min$^{-1}$ load speed. Hydrogel specimens were evaluated in a fiber shape ( ~ 0.4 mm diameter and ~50 mm length, and ~10 mm gauge length at the machine).

### Preparation of artificial ligaments comprising multiple PVA−NC fibers

2, 4, 8, and 50 PVA−NC fibers obtained after the FT process were parallelly aligned at the same length of approximately 10 cm. The aligned fibers were then twisted to obtain one braided hydrogel bundle. The two ends of the braided hydrogel bundle were clipped to prevent untied and then submerged in 2.8 M Na$_3$Citrate solution for 24 h.

### Characterization

X-ray diffraction (XRD) analysis was performed using Cu X-rays (λ = 1.54 Å) at 2θ = 10–80° in a step of 0.02° with 0.5 s dwell (Bruker D8 Advanced equipped with a LynxEye CCD detector). The morphology of hydrogels and NC powder was characterized by the scanning electron microscope (SEM, FEI Quanta FEG 250). In Differential Scanning Calorimetry (DSC) measurement, samples were subsequently placed in Tzero pans and heated up from 50 °C to 260 °C at a rate of 20 °C min$^{-1}$ under a nitrogen atmosphere with a flow rate of 40 mL min$^{-1}$ (TA Instrument Discovery SDT 650). The Fourier transform infrared (FTIR) spectra were taken by a Nicolet™ iS50 Fourier transform infrared (FTIR) spectrometer, fitted with Smart-iTR™ diamond Attenuated Total Reflectance (ATR) attachment, from 4000 to 500 cm$^{-1}$ at a resolution of 4 cm$^{-1}$ with a total of 32 scans under 23 °C. For SEM, XRD, DSC, and FTIR analyses, the lyophilized samples were used. To prepare these samples, all hydrogels were first rinsed with water for 72 h to remove salt ions that recrystallize during the lyophilization; water was frequently replaced. Particularly for XRD, DSC, and FTIR analyses, the water-rinsed hydrogels were treated with glutaraldehyde and hydrochloric acid to fix PVA crystallinity and network structure before drying; the hydrogels were submerged in an aqueous solution composed of 10 mL glutaraldehyde, 500 µL hydrochloric acid, and 100 mL water for 6 h. The obtained hydrogels were rinsed with water for 24 h to remove residual glutaraldehyde and

hydrochloric acid in the hydrogel matrix. The resulting hydrogels were submerged in liquid nitrogen for immediate freezing, and then the frozen samples were dried in a lyophilizer at −50 °C for 72 h (Labconco FreeZone).

The water content of the hydrogels is measured by the difference between their weights before and after drying. The weight of the hydrogel before drying ($m_O$) was measured once the superficial water was wiped, and the weight of the hydrogel after drying ($m_d$) was measured once dehydrated in an oven at 70 °C for 72 h. The water content was calculated as $[(m_O − m_d)/m_O] × 100\%$. The swelling ratio of the hydrogel was measured by the difference between its weights before and after reaching the equilibrium state in water. From the weight before swelling ($m_O$) and after swelling ($m_s$), the swelling ratio was calculated as $[(m_s − m_O)/m_O] × 100\%$.

For PVA−NC hydrogels with 3D complex structures, the PVA−NC mixture with 8 wt% NC was printed into pre-designed structures using a 3D printer (ROKIT INVIVO or CELLINK BIO X). The printed structures were then treated with the freezing-thawing and salting-out processes.

The real-time resistance changes of the hydrogel under different deformations were recorded by a source meter station (2450 Source-Meter, KEITHLEY) at 10 mA current. $R_O$ denotes the initial resistance of the hydrogel, and $\Delta R$ represents the changes in resistance, calculated as the difference between the resistance in the deformed state and the initial state ($\Delta R = R − R_O$).

## Data availability
Data is available from the corresponding author on request. Source data are provided with this paper.

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

## Acknowledgments
This research was supported by the National Research Foundation of Korea (NRF) grant funded by the Ministry of Science and ICT (MSIT), RS-2025-00557115 (J.B.) and the NRF Brain Pool program funded by MSIT, RS-2024-00408466 (D.J.).

## Author contributions
Conceptualization: D.J., D.S., and J.B.; Methodology and investigation: D.S., D.J., and J.B.; Writing: D.J., D.S., and J.B.; Supervision: J.B.

## Competing interests
The authors declare no competing interests.
