## [Transparent Peer Review file · Nature Communications]

Hierarchically structuralized hydrogels with ligament-like mechanical performance

Corresponding Author: Professor Jinhye Bae

Version 0:

Reviewer comments:

Reviewer #1

(Remarks to the Author)

The manuscript presents a novel fabrication strategy for polyvinyl alcohol-nanoclay (PVA-NC) hydrogel fibers with hierarchical structures mimicking natural ligaments. The hydrogels achieve impressive mechanical properties (tensile strength: 61 ± 8 MPa, elastic modulus: 131 ± 15 MPa, toughness: 135 ± 11 MJ m⁻³) and demonstrate potential as artificial ligaments and strain sensors. While the work is technically sound and the data well-supported, the advancement beyond existing literature is insufficiently articulated. Key concerns include the incremental nature of the freeze-thaw-salting out (FTS) method, well-established biomimetic concepts, limitations in 3D printing precision, and insufficient novelty in nanoparticle reinforcement. Therefore, I am not convinced to support the publication of this manuscript in Nature Communications. Even if all my concerns are addressed, I am not sure this journal is the best outlet for this work.

Major Concerns

1. Limited Novelty of the Core Fabrication Method

The FTS strategy (freeze-thaw followed by salting-out) is heavily inspired by prior work (e.g., Hua et al., Nature 2021, 590:594–599), which already demonstrated ultra-strong hydrogels via similar mechanisms.

While the addition of nanoclay (NC) and shear-induced alignment during extrusion are noted, the manuscript fails to:

Quantify how much NC improves properties beyond the FTS baseline (e.g., via controls without NC).
Explain why NC-driven agglomeration is superior to other nanoparticle systems (e.g., silica, cellulose nanocrystals) extensively studied for hydrogel reinforcement (e.g., Gaharwar et al., Soft Matter 2010, 6:2364; Koetting et al., J. Control. Release 2015, 130:246).

2. Overlap with Existing Biomimetic Ligament Designs

The concept of "biomimetic hierarchical bundles" for ligament repair is well-explored. Recent studies (e.g., Wang et al., Adv. Funct. Mater. 2024, 35:2415737; Screen et al., J. Biomech. 2011, 44:694) have replicated collagen fiber bundling, mechanical anisotropy, and load-sharing mechanisms.

The 50-fiber bundle lifting 13.6 kg (Fig. 4e) is compelling but lacks comparison to state-of-the-art artificial ligaments (e.g., carbon nanotube/polymer composites).

3. Inadequate Characterization of 3D Printing Limitations

The extrusion-based 3D printing (600- μ m nozzle) produces fibers with ~450–500 μ m diameters (Fig. 1f). This resolution is notably coarser than cutting-edge bioprinting techniques (e.g., voxelated hydrogels at 10 μ m scale; Skylar-Scott et al., Nature 2024, 631:783).

The manuscript claims "versatility for customized 3D structures" (Suppl. Movie 4) but only demonstrates simple fibers/bundles. Complex architectures (e.g., porous scaffolds, anatomical shapes) are absent.

4. Superficial Treatment of Nanoparticle Reinforcement Mechanisms

The role of NC as a "mechanical reinforcement and rheological modifier" (Fig. 2b–d) is well-documented for clay-polymer hydrogels (e.g., Haraguchi et al., Adv. Mater. 2002, 14:1120).

Claims of "unique" load-transfer via PVA-NC agglomerates (Fig. 3i) lack novelty compared to: Classical theories of filler-matrix stress transfer (e.g., shear-lag model).

Recent studies on mineralized collagen analogs (Ping et al., Science 2022, 376:188).

Minor Concerns

1. Statistical Rigor: Mechanical property data (e.g., ± 8 MPa for strength) should specify sample size ($n \geq 5$ recommended).

2. Biological Relevance: Ligament-matching mechanics are highlighted, but cytocompatibility, degradation, or in vivo safety data are absent.

3. Strain Sensing: The $\Delta R/R_0$ response (Fig. 4g,i) lacks quantification of sensitivity/gauge factor.

Reviewer #2

(Remarks to the Author)

This work integrates poly(vinyl alcohol) (PVA) with nanoclay (NC) and employs a combination of shear-induced alignment, freezing-thawing, and salting-out strategy to fabricate hydrogel fibers. The hydrogel fibers exhibit excellent mechanical properties, with high strength (61 ± 8 MPa), stiffness (131 ± 15 MPa), and toughness (135 ± 11 MJ/m³). The fiber bundles mimic the structure of ligaments and show enhanced load-bearing capacity. The interaction between PVA matrix and NC is revealed by XRD, FTIR and DSC. Overall, this is a well-executed and impactful study with strong mechanical and structural results. One minor suggestion is to provide the water content and swelling ratio of the resulting fibers to understand the network structure and hydration behavior.

Reviewer #3

(Remarks to the Author)

1. The format of references is not uniform.
2. In Fig. 4a, I cannot understand why the two samples varied so much in the size and which is the braided bundle.
3. I cannot what is the meaning of "the melting temperature became ill-defined". The authors should provide more explanation for this.
4. The authors should provide more details for the formation of braided hydrogel fiber bundles.
5. The preparation of hydrogel is too simple.
6. We recommend the authors to pay attention to some research work, such as Chemical Engineering Journal, 2023, 463, 142414.
7. "ml min⁻¹" should be changed into "mL min⁻¹".
8. The illustration in Fig.2a is not clearly. I cannot find the specific interaction between the PVA and NC.
9. Other characterizations such as stability of hydrogel fibers at various liquid phase environments should be performed.
10. Some characterizations for Nanoclay should be provided.

Version 1:

Reviewer comments:

Reviewer #2

(Remarks to the Author)

The authors have addressed my concerns. I recommend it to be published in Nature Communications.

Reviewer #3

(Remarks to the Author)

[Note from the Editor: reviewer 3 also assessed the responses to reviewer 1]

Basically I think the authors have revised the manuscript according to the reviewers' comments and I agree the publication of this manuscript.

Responses to Reviewers' Comments

The following is the Response to the Reviewers' Comments for the manuscript entitled "**Biomimetic hierarchically structuralized hydrogels with superior strength, stiffness, and toughness resembling natural ligaments**" (NCOMMS-25-35252) to *Nature Communications*.

Reviewer #1

General Comment: The manuscript presents a novel fabrication strategy for polyvinyl alcohol-nanoclay (PVA-NC) hydrogel fibers with hierarchical structures mimicking natural ligaments. The hydrogels achieve impressive mechanical properties (tensile strength: 61 ± 8 MPa, elastic modulus: 131 ± 15 MPa, toughness: 135 ± 11 MJ m⁻³) and demonstrate potential as artificial ligaments and strain sensors. While the work is technically sound and the data well-supported, the advancement beyond existing literature is insufficiently articulated. Key concerns include the incremental nature of the freeze-thaw-salting out (FTS) method, well-established biomimetic concepts, limitations in 3D printing precision, and insufficient novelty in nanoparticle reinforcement. Therefore, I am not convinced to support the publication of this manuscript in *Nature Communications*. Even if all my concerns are addressed, I am not sure this journal is the best outlet for this work.

Response: We appreciate the Reviewer's thoughtful comments. In response to the concerns raised, we have clarified the novelty of this study, particularly, the fabrication method, the rationale for selecting polymeric and inorganic components, and the distinguishing features of our work, including exceptional mechanical properties and broad applicability of the developed hydrogels, compared to previously reported studies. These clarifications have been incorporated into the revised manuscript. The added/revised texts and figures in the revised manuscript have been highlighted in yellow. The detailed point-by-point responses to each comment are provided as follows.

Specific Comments

#1. Limited Novelty of the Core Fabrication Method. The FTS strategy (freeze-thaw followed by salting-out) is heavily inspired by prior work (e.g., Hua et al., *Nature* 2021, 590:594–599), which already demonstrated ultra-strong hydrogels via similar mechanisms. While the addition of nanoclay (NC) and shear-induced alignment during extrusion are noted, the manuscript fails to: Quantify how much NC improves properties beyond the FTS baseline (e.g., via controls without NC). Explain why NC-driven agglomeration is superior to other nanoparticle systems (e.g., silica, cellulose nanocrystals) extensively studied for hydrogel reinforcement (e.g., Gaharwar et al., *Soft Matter* 2010, 6:2364; Koetting et al., *J. Control. Release* 2015, 130:246).

Response: Thank you for your valuable comment. Although several fiber-shaped or complex-shaped hydrogels have been reported in previous studies, they have not demonstrated excellent mechanical properties (i.e., the combination of strength > 50 MPa, elastic modulus > 100 MPa, toughness > 100 MJ m⁻³, stretchability $> 300\%$). In this work, we overcome these limitations,

achieving hydrogel fibers with ligament-level strength and stiffness along with outstanding toughness and stretchability through an integrated fabrication and reinforcement strategy.

To fabricate fiber-shaped hydrogels, we first engineered the PVA-based precursor mixture to behave as an extrudable viscoelastic solid. This means that the precursor must possess high viscosity and storage modulus surpassing loss modulus ($G' \gg G''$). For that, we utilized nanoclay (NC), a sort of colloidal particle, as a rheological modifier [R1]. Once NC was added to a PVA solution, the PVA-NC mixture became a viscoelastic solid, demonstrated as increases in viscosity and G' surpassing G'' (Figures 2b and 2c). Without NC, the pure PVA solution was liquid-like state with significantly low viscosity and $G'' \gg G'$. Among different NC content in PVA-NC mixture, 8 wt% NC was the best condition for extruding PVA-NC mixture and for shape retention once extruded (Figure 2d). The extruded fiber-shaped mixtures were subsequently treated using freezing-thawing (FT) and salting-out (S) processes to produce strong, stiff, and tough hydrogel fibers (Figures 1f and 2a).

From the perspective of fiber fabrication, 8% NC was required. In other words, pure PVA solution and PVA-NC mixture with low NC content could not be used for the fiber fabrication; therefore, in the manuscript, there was no mechanical testing results for PVA-NC/FT/S hydrogels with different NC contents. Instead, in the manuscript, we systemically investigated the effect of the number of FT cycles (Figures 2e–h) and the concentration of salt ions during the S process (Figures 3a–c) on the mechanical properties of hydrogel fibers. Also, by comparing the mechanical properties of previously reported PVA hydrogels treated by an FT, S, or both processes, we were clearly able to notice the mechanical reinforcing effect of NC on the PVA hydrogel (Figures 1g and 1h, and Supplementary Table 1). Although we basically employed the FTS strategy, the integration of hydrogel fiber fabrication (along with proper component selection) and FTS processes has not been tried, so the systematic investigation of the mechanical properties and the consequential understanding must be worthwhile.

The superiority of NC from the perspective of mechanical enhancement is also related to the nature of NC. Since NC is a sort of colloidal particle as mentioned above and has good affinity with polyvinyl alcohol (PVA) [R1, R2], we expected that NC would agglomerate within the PVA-based hydrogel matrix (rather than solely separated from the matrix) during FT and S processes. Indeed, the PVA-NC/FT/S hydrogel displayed a distinct internal structure where large-sized PVA-NC hybrid agglomerates are incorporated in the continuous, interconnected PVA matrix (Figure 1e), which is barely observed in other composite hydrogels. The formation of PVA-NC agglomerates was further supported through FTIR and DSC results (Figures 3g and 3h). Consequently, the PVA-NC/FT/S hydrogel demonstrated outstanding mechanical properties, tensile strength of 61 ± 8 MPa, elastic modulus of 131 ± 15 MPa, and toughness of 135 ± 11 MJ m⁻³, which are significantly superior to those of previously reported hydrogels (Figures 1g and 1h, and Supplementary Table 1). Notably, the mechanical enhancement in strength and stiffness did not compromise stretchability, and thus, the hydrogel fibers demonstrated outstanding toughness. If NC were simply dispersed without forming hybrid agglomerates, load transfer and toughness enhancement would be substantially limited. These reinforcement mechanisms differ from those of conventional inorganic-filler-reinforced composites. Inorganic fillers typically strengthen and stiffen materials via stress transfer

described by shear-lag model but usually restrict toughness and stretchability. Instead, the district structure of our hydrogel and its mechanical reinforcement mechanisms (i.e., PVA-NC hybrid agglomerates incorporated in the continuous, interconnected matrix) are similar to those of partially mineralized tissues—an intermediate between unmineralized tissue, such as tendon, and fully mineralized tissue, such as bone—which exhibit high strength, stiffness, and toughness beyond predictions of classical theory like the shear-lag model.

[R1] Cummins, H. Liquid, glass, gel: The phases of colloidal Laponite. *Journal of Non-Crystalline Solids*. **353**, 3891-3905 (2007).

[R2] Das, P., *et al.* Nacre-mimetics with synthetic nanoclays up to ultrahigh aspect ratios. *Nat. Commun.* **6**, 5967 (2015)

These descriptions are provided in the manuscript, with certain sections (highlighted in yellow) revised to more clearly emphasize the distinguishing features of our work.

(Revised manuscript, Abstract) Mechanical properties of synthetic hydrogels remain inferior to those of load-bearing tissues such as ligaments, one of the strongest and stiffest natural hydrogels in the human body. Inspired by biological structures and their mechanisms conferring exceptional mechanical properties, we report strong, stiff, and tough hydrogels composed of fiber-shaped elements that can be assembled into parallel bundles, closely resembling natural ligaments. These hydrogel fibers, readily fabricated with diameters of a few hundred micrometers, comprise polymer–particle hybrid agglomerates embedded in a continuous, interconnected polymer matrix. Strong polymer–particle interactions combined with spatial confinement within the agglomerates enable efficient load transfer, resulting in significant load-transfer lengths and substantial energy dissipation across the network. This design overcomes the conventional trade-offs between strength/stiffness and toughness/stretchability in polymer composites, thereby achieving a tensile strength of 61 ± 8 MPa, elastic modulus of 131 ± 15 MPa, toughness 135 ± 11 MJ m⁻³, and stretchability exceeding 400%. These properties surpass those of previously reported synthetic hydrogels and approach those of natural ligaments. When assembled into millimeter-scale hierarchical bundles, the hydrogels mimic the structural organization of ligaments, sustain loads of tens of kilograms, and function as strain sensors.

(Revised manuscript, Pages 5–6) To fabricate strong, stiff, and tough hydrogel fibers, we employed a combination of PVA and NC with freezing-thawing (FT) and salting-out (S) processes (Fig. 2a). In conventional methods, liquid-like PVA solutions should be directly injected into a coagulation bath (e.g., high-concentration salt solution) to form PVA networks solely via the S process without FT process, resulting in solidified hydrogel fibers^{21,23,31}. By contrast, the viscoelastic PVA-NC mixture was extrudable into long fiber shapes without a coagulation bath because NC served as both a mechanical reinforcement and rheological modifier imparting a viscoelastic solid-state property³². During extrusion, the PVA and NC components were subjected to shear-induced alignment (Fig. 1d)^{21,33}. Following extrusion, the PVA-NC fibers were treated with sequential FT and S processes, resulting in PVA-NC/FT/S hydrogel fibers with 450–500 μm diameters using a 600 μm nozzle. During the FT process, NC particles agglomerated, forming hydrogels with PVA-NC hybrid agglomerates and continuous polymer matrices (Fig. 1e). We expected that the combination of sequential FT and

S treatments forming a distinct structure, composed of PVA-NC hybrid agglomerates within the continuous polymer matrix with aligned polymer chains, would enhance the mechanical properties of hydrogels^{28,34-38}.

To confirm the effect of NC on PVA solution and access the potential for fiber formation, we first investigated the rheological properties of PVA-NC mixtures with varying NC content (2–8 wt%). Unlike pure PVA solution, which exhibits a greater loss modulus (G'') than storage modulus (G'), the PVA-NC mixtures demonstrated a viscoelastic solid behavior with G' exceeding G'' in the absence of external force (Fig. 2b). When subjected to external force exceeding a critical level, the mixtures underwent shear-yielding, suggesting their good extrudability^{33,39}. These mixtures also exhibited the shear-thinning behavior (Fig. 2c). While the PVA-NC mixture has high viscosity at low shear rates (i.e., before and after extrusion), the mixture temporarily has low viscosity under high shear rates during extrusion. Among the tested formulations, the PVA-NC mixture with 8 wt% NC demonstrated the most appropriate viscosity and viscoelastic properties, facilitating smooth extrusion through syringe nozzles and retaining its shape after extruded (Fig. 2d). Mixtures with less than 8 wt% NC lost their shape in 5 minutes after extruded (Fig. 2d, inset photographs), while mixtures with more than 8 wt% NC were too thick to form long fibers through continuous extrusion.

Using the PVA-NC mixture with 8 wt% NC content, we fabricated PVA-NC hydrogel fibers treated with various FT cycles and different salt types to evaluate the effect of FT and S processes on mechanical properties. Sodium sulfate (Na_2SO_4) and sodium citrate ($\text{Na}_3\text{Citrate}$), known for their strong salting-out effect¹⁵, were used at a fixed concentration of 1.5 M in this evaluation. PVA-NC/S hydrogel fibers, treated only with salting-out directly without FT cycles (i.e., 0 cycle), were significantly weaker and softer than the PVA-NC/FT/S hydrogel fibers (Fig. 2e and 2f, black lines). This finding demonstrates that salting-out alone has a limitation in achieving superior mechanical properties. Overall mechanical properties (fracture strength and strain, and elastic modulus) of PVA-NC/FT/S hydrogel fibers remained generally consistent across one to seven FT cycles. However, they were significantly affected by the salt type due to variations in salting-out efficiencies¹⁵ (Figs. 2e and 2f). Specifically, $\text{Na}_3\text{Citrate}$ -treated hydrogel fibers exhibited significantly higher fracture strength and elastic modulus compared to Na_2SO_4 -treated hydrogel fibers (Supplementary Fig. 1); thus, $\text{Na}_3\text{Citrate}$ was the most effective salts conferring high mechanical properties.

Conversely, PVA-NC/FT hydrogel fibers fabricated without salting-out exhibited lower mechanical strength, stiffness, and stretchability (Fig. 2g). While increasing the number of FT cycles slightly improved stretchability, the mechanical properties of PVA-NC/FT fibers were an order of magnitude inferior to those of the PVA-NC/FT/S fibers. These results underscore the importance of the sequential combination of FT and S processes in achieving enhanced mechanical properties. Notably, even a single FT cycle followed by $\text{Na}_3\text{Citrate}$ treatment is both efficient and effective in producing mechanically robust hydrogel fibers with a tensile strength of 46 MPa and an elastic modulus of 91 MPa (Fig. 2h). **Because pure PVA solution and PVA-NC mixtures with a low NC content could not be used for the fiber fabrication**, subsequent experiments focused on evaluating the mechanical properties and mechanisms of hydrogel fibers, **PVA-NC/FT/S at the fixed 8 wt% NC**, treated with varying $\text{Na}_3\text{Citrate}$ concentrations after one FT cycle.

Enhanced mechanical properties and reinforcing mechanisms. We evaluated the effect of Na₃Citrate concentration (0.1–2.8 M) on the mechanical properties and achieved a combination of high strength, stiffness, and toughness, comparable to natural ligaments (Fig. 3a–c). As the Na₃Citrate concentration increased, the mechanical properties of the PVA-NC/FT/S hydrogel fibers improved proportionally.

(Revised manuscript, Pages 7–9) Moreover, Fourier transform infrared (FTIR) spectroscopy analysis demonstrated the interactions between PVA and NC (Fig. 3g). Compared to PVA/FT/S, PVA-NC/FT/S exhibited an amplified peak intensity at 3400 cm⁻¹, indicating increased O–H stretching vibrations^{16,64}. While NC displayed two prominent peaks at ~3600 and 3400 cm⁻¹, the 3600 cm⁻¹ peak was absent in the PVA-NC/FT/S hydrogel, suggesting that 3400 cm⁻¹ peak predominantly arises from interactions between PVA and NC. This interaction likely results from the agglomeration of PVA and NC, which expels water molecules previously bound to hydrated PVA chains, thereby revealing hidden O–H stretching vibrations. Additionally, the slight increase in peak intensity at a lower wavenumber (~3170 cm⁻¹) implies weakened O–H stretching due to the formation of hydrogen bonding between PVA and NC^{65,66}. Differential scanning calorimetry (DSC) analysis further confirmed substantial binding interactions between PVA and NC (Fig. 3h). As the NC content increased from 0 to 8 wt%, the PVA melting temperature gradually shifted upward, accompanied by a reduction in melting peak sharpness and intensity. At 8 wt% NC, the melting peak **was significantly broadened and not precisely defined**, reflecting significantly restricted thermal motion of PVA chains. This reduction in chain mobility was attributed to the spatial confinement of PVA chains between NC, a phenomenon intensified in the PVA-NC agglomerates (Fig. 1e)^{67–69}.

Based on these results, including FTIR, DSC, and SEM analyses, the mechanical reinforcement and fracture mechanisms of the PVA-NC/FT/S hydrogels are proposed as follows (Fig. 3i). Upon tensile loading, the applied stress is effectively transferred from the flexible PVA matrix to the rigid PVA-NC agglomerates through strong cohesive interactions, including hydrogen bonding and spatial confinement between PVA chains and NC particles^{36–38,70}. **Because the PVA-NC hybrid agglomerates are intertwined within the continuous PVA matrix and the PVA chains are spatially connected across the agglomerates and matrices, stretching of PVA chains under the loading accompanies the deformation of agglomerates.** Over these extended load-transfer lengths, substantial energy dissipation occurs^{29,30,36,37}, resulting in the hydrogel's high fracture strength and strain (Fig. 3a). With further stretching, microcracks are likely to initiate within the PVA matrix and propagate along the agglomerate surfaces, delaying ultimate failure^{36,37}. This sequential fracture mechanism imparts significant tensile strength and elastic modulus while impeding failure and enhancing toughness, overcoming the conventional trade-off between strength/stiffness and toughness/stretchability.

Importantly, these reinforcement mechanisms differ from those of conventional inorganic platelet-reinforced composites. In typical composites, inorganic platelets enhance strength and stiffness primarily through load transfer described by the shear-lag model, where applied stress is predominantly transferred from the polymer matrix to individual reinforcements via interfacial shear^{67,71–73}. Under this mechanism, the energy dissipation zone (or process/bridging/damage zone) is inherently limited, resulting in restricted toughness and stretchability^{67,74–76}. By contrast, the PVA-NC/FT/S hydrogels feature polymer–particle hybrid

agglomerates embedded within a continuous polymer matrix, where polymer chains are spatially interconnected across the agglomerates and matrices throughout the entire hydrogel. This distinct structure enables effective load transfer across long distances, which leads to exceptional toughness and stretchability. If NC were only individually dispersed or poorly integrated in the matrix, load transfer would be confined to short distances, substantially diminishing mechanical reinforcement and, in particular, toughness. This structural design resembles that of partially mineralized tissues—an intermediate between unmineralized tissue (e.g., tendon) and fully mineralized tissue (e.g., bone)—which are known to achieve high strength, stiffness, and toughness beyond predictions of classical theories like the shear-lag model⁷⁷⁻⁷⁹. Therefore, the distinct internal structure enabled our hydrogel to simultaneously achieve high strength, stiffness, toughness, and stretchability, breaking the traditional trade-off observed in polymer composites.

(Revised manuscript, Page 11) During these processes, NC did not simply separate from the matrix but instead aggregated within the PVA matrix, forming PVA-NC hybrid agglomerates incorporated in the continuous, interconnected PVA matrix. The applied load was effectively transferred to these agglomerates through strong cohesive interactions and spatial confinement between PVA chains and NC particles, enabling substantial energy dissipation over extended load-transfer lengths. Therefore, the PVA-NC/FT/S hydrogel fibers demonstrated outstanding tensile strength (61 ± 8 MPa), elastic modulus (131 ± 15 MPa), and toughness (135 ± 11 MJ m⁻³), resembling natural ligaments.

(Added references)

71. Das, P. et al. Nacre-mimetics with synthetic nanoclays up to ultrahigh aspect ratios. *Nat. Commun.* **6**, 5967 (2015).
72. Gao, H. Application of Fracture Mechanics Concepts to Hierarchical Biomechanics of Bone and Bone-like Materials. *Inter. J. Fract.* **138**, 101-137 (2006).
73. Ji, D., Nguyen, T. L. & Kim, J. Bioinspired Structural Composite Hydrogels with a Combination of High Strength, Stiffness, and Toughness. *Adv. Funct. Mater.* **31**, 2101095 (2021).
74. Ducrot, E., Chen, Y., Bulters, M., Sijbesma, R. P. & Creton, C. Toughening Elastomers with Sacrificial Bonds and Watching Them Break. *Science* **344**, 186-189 (2014).
75. Tang, J., Chen, X., Liu, F., Suo, Z. & Tang, J. Why are soft collagenous tissues so tough? *Sci. Adv.* **11**. eadw0808 (2025).
76. Ritchie, R. O. The conflicts between strength and toughness. *Nat. Mater.* **10**, 817-822 (2011).
77. Golman, M. et al. Toughening mechanisms for the attachment of architected materials: The mechanics of the tendon enthesis. *Sci. Adv.* **7**. eabi5584 (2021).
78. Alcântara, A. et al. Molecular-Scale Interactions at Mineralized Collagen Interfaces Prevent Network Percolation, Preserving Compliance. *ACS. Nano* **19**, 31350-31362 (2025).
79. Seknazi, E. & Pokroy, B. Residual Strain and Stress in Biocrystals. *Adv. Mater.* **30**, e1707263 (2018).

#2. Overlap with Existing Biomimetic Ligament Designs. The concept of "biomimetic hierarchical bundles" for ligament repair is well-explored. Recent studies (e.g., Wang et al., *Adv. Funct. Mater.* 2024, 35:2415737; Screen et al., *J. Biomech.* 2011, 44:694) have replicated collagen fiber bundling, mechanical anisotropy, and load-sharing mechanisms. The 50-fiber bundle lifting 13.6 kg (Fig. 4e) is compelling but lacks comparison to state-of-the-art artificial ligaments (e.g., carbon nanotube/polymer composites).

Response: Thank you for raising this important point. As we are also aware of recent studies on artificial ligaments, we would like to highlight how our work differs from and advances beyond these studies. Unlike previous reports, our work introduces a novel class of fiber-shaped hydrogels—PVA-NC/FT/S hydrogel fibers—with a unique internal structure and exceptional mechanical performance at the single-fiber material level. This study is the first demonstration of such hydrogel fibers achieving simultaneous high strength, stiffness, toughness, and stretchability, enabled by a rationally designed hierarchical structure and polymer–particle hybridization strategy.

The hydrogel fibers, with diameters of a few hundred micrometers, were readily fabricated and feature polymer–particle hybrid agglomerates intertwined within a continuous polymer matrix. Polymer chains are spatially connected across the agglomerates and matrix domains throughout the hydrogel. This distinct architecture enabled effective long-length load transfer, thereby resulting in exceptional mechanical performance, overcoming the traditional trade-off in polymer composites. Specifically, we achieved a tensile strength of 61 ± 8 MPa, elastic modulus of 131 ± 15 MPa, toughness of 135 ± 11 MJ m⁻³, and stretchability over 400%, a performance combination not previously reported in hydrogels.

Building on this material innovation, we further fabricated artificial ligaments by braiding multiple hydrogel fibers. Owing to the simplicity of the fabrication process (no special instruments, harmful chemicals, etc.), scalability, and robustness of individual fibers, we readily produced hydrogel bundles in varying sizes. Notably, a braided bundle with multiple fibers successfully lifted 13.6 kg without rupture, demonstrating excellent load-bearing capability.

Most prior artificial ligament studies focused on dry materials rather than hydrogel materials. For example, a recent study (Hierarchical helical carbon nanotube fibre as a bone-integrating anterior cruciate ligament replacement. *Nat. Nanotechnol.* 2023, 18, 1085–1093) presented an artificial ligament composed of carbon nanotube fibers, not hydrogel-based materials.

A few studies have reported hydrogels with high strength and stiffness, such as “Multifunctional tendon-mimetic hydrogels, *Sci. Adv.* 2023, 9, eade6973 & Super Strong and Tough PVA Hydrogel Fibers Based on an Ordered-to-Disordered Structural Construction Strategy Targeting Artificial Ligaments. *Adv. Funct. Mater.* 2025, 35, 2415737”. While these hydrogels achieved ligament-level strength and stiffness, their stretchability remained highly limited (only a few tens of percent). This result implies that high strength and stiffness were achieved at the expense of stretchability and toughness. Notably, the hydrogels in the latter study (*Adv. Funct. Mater.* 2025, 35, 2415737) contained substantially lower water content (20–30 wt%) than both previously reported enhanced hydrogels (60–90wt%) and natural load-bearing tissues (i.e., natural hydrogels), such as ligament and tendon (50–70 wt%). In contrast, the PVA-NC/FT/S hydrogel in our study maintains a water content of 56 wt%, closely matching that of natural load-bearing tissues (Supplementary Fig. 3), while simultaneously achieving high strength, stiffness, toughness, and stretchability.

These previous studies relied heavily on a drawing process (stretching + drying + annealing) to achieve polymer chain alignment and reinforcement, a strategy known to

significantly stiffen polymeric materials. Therefore, during the revision period, we similarly employed stretching and dry-annealing processes on our hydrogel fibers and confirmed the further enhancement in the mechanical performance (Figs. 3j–3m). The resulting hydrogel fibers exhibited tensile strength of 62–79 MPa and elastic modulus of 511–1182 MPa, while still maintaining good stretchability over 150% deformation. Therefore, superior mechanical performance of our PVA-NC-based hydrogels demonstrates that our design strategy achieves a balance of strength, stiffness, and toughness, distinguishing it from previously reported hydrogels.

Supplementary Fig. 3. Water content of two different hydrogels. Error bars correspond to standard deviations at $n \geq 5$.

(Revised manuscript, Experimental section) The water content of the hydrogels is measured by the difference between their weights before and after drying. The weight of the hydrogel before drying (m_0) was measured once the superficial water was wiped, and the weight of the hydrogel after drying (m_d) was measured once dehydrated in an oven at 70 °C for 72 h. The water content was calculated as $[(m_0 - m_d)/m_0] \times 100\%$.

Fig. 3 (j) Stress-strain curves, (k) tensile strengths, (l) elastic moduli, and (m) toughness of PVA-NC/FT, PVA-NC/FT/St/S and PVA-NC/FT/St-D-A/S hydrogel fibers ($n \geq 4$, error bars correspond to standard deviations).

(Revised manuscript, Page 9) Furthermore, we achieved additional improvements in the mechanical performance of the hydrogel, particularly in strength and stiffness (Figs. 3j–3m), by integrating previously established stretching and dry-annealing processes^{41,80–82}. Specifically, the PVA-NC/FT hydrogel fiber was first stretched to 100% deformation; the fracture elongation of PVA-NC/FT was approximately 150–160% (Figure 2g). This stretched sample (PVA-NC/FT/St) was then immersed in 2.8 M Na₃Citrate solution for the salting-out process, yielding a PVA-NC/FT/St/S hydrogel fiber. For a PVA-NC/FT/St-D-A/S hydrogel fiber, the stretched sample was additionally dried and annealed at 70 °C for one day before the salting-out process. As a result of polymer chain alignment and densification induced by the stretching and dry-annealing processes^{41,64,73,80,82}, both PVA-NC/FT/St/S and PVA-NC/FT/St-D-A/S hydrogel fibers exhibited exceptional mechanical properties (Figs. 3j–3m). Their elastic moduli reached 511 MPa and 1182 MPa, respectively, while still maintaining good stretchability beyond 150% deformation. Notably, for the PVA-NC/FT/St-D-A/S hydrogel, fracture strength, elastic modulus, and toughness are 40.2-, 1225-, and 61.2-fold higher, respectively, than those of the PVA-NC/FT hydrogel.

(Revised manuscript, Page 11) In addition, by integrating stretching and dry-annealing processes to enhance polymer chain alignment and densification, the hydrogel fibers achieved elastic modulus in the range of several hundred to over one thousand MPa, while still maintaining good stretchability beyond 150% deformation.

(Revised manuscript, Experimental section) The extruded PVA-NC mixture underwent freezing (8 h)–thawing (8 h) for 1–5 cycles (Fig. 2a, freezing-thawing process for PVA-NC/FT hydrogels), and subsequently was immersed in a salt solution for the salting-out treatment for over 24 h (Fig. 2a, salting-out process for PVA-NC/FT/S hydrogels). For the PVA-NC/FT/St/S hydrogels, the PVA-NC/FT hydrogel was first stretched to a 100% deformation (i.e., stretch process), the ends of stretched hydrogel were fixed to maintain the extended length. The sample was then immersed in a salt solution for over 24 hours as the salting-out treatment. In the case of PVA-NC/FT/St-D-A/S hydrogels, the PVA-NC/FT hydrogel was first stretched in a 100 % deformation (i.e., stretch process), the ends of stretched hydrogel were fixed, and then the stretched hydrogel was dried in an oven at 70 °C for a day (i.e., dry-annealing process); subsequently, the dried sample was immersed in a salt solution for the salting-out treatment for over 24 h (i.e., salting-out treatment).

(Added references)

81. Sun, M. et al. Multifunctional tendon-mimetic hydrogels. *Sci. Adv.* **9**, eade6973 (2023).
82. Mredha, M. et al. A Facile Method to Fabricate Anisotropic Hydrogels with Perfectly Aligned Hierarchical Fibrous Structures. *Adv. Mater.* **30**, 1704937 (2018).

#3. Inadequate Characterization of 3D Printing Limitations. The extrusion-based 3D printing (600- μ m nozzle) produces fibers with \sim 450–500 μ m diameters (Fig. 1f). This resolution is notably coarser than cutting-edge bioprinting techniques (e.g., voxelated hydrogels at 10 μ m scale; Skylar-Scott et al., *Nature* 2024, 631:783). The manuscript claims "versatility for customized 3D structures" (Suppl. Movie 4) but only demonstrates simple fibers/bundles. Complex architectures (e.g., porous scaffolds, anatomical shapes) are absent.

Response: Thanks for the Reviewer's comment. In the original manuscript, we demonstrated a printed rectangular lattice structure (Supplementary Fig. 6), supporting the claim that "the PVA-NC mixture was printable in diverse shapes beyond the fiber form."

Supplementary Fig. 5. Photographs of the printed structure of PVA-NC/FT/S hydrogel.

To further support the ability to create customized printed hydrogel structures, we have added new results in the revised manuscript (Supplementary Fig. 6) showing different printed structures with increased structures' height.

Supplementary Fig. 6. Photographs of the printed structures of PVA-NC/FT/S hydrogel. (a) Square structure printed by a grid printing route. (b) Hexagon structure printed by a honeycomb printing route.

Achieving precise printing at the tens-of-micrometer scale (instead of the 500–600 μm resolution of single extrusion) would require substantial fine-tuning in both PVA and NC concentrations to accommodate printing with a ten-micrometer nozzle, which lies beyond the primary focus of this study. Since our goal in this study is to develop macro-scale fiber-shaped hydrogels with superior mechanical properties, future work may focus on fabricating hydrogels with fine structural features at the tens-of-micrometer scale to advance state-of-the-art hydrogel printing techniques.

#4. Superficial Treatment of Nanoparticle Reinforcement Mechanisms. The role of NC as a "mechanical reinforcement and rheological modifier" (Fig. 2b–d) is well-documented for clay-

polymer hydrogels (e.g., Haraguchi et al., *Adv. Mater.* 2002, 14:1120). Claims of "unique" load-transfer via PVA-NC agglomerates (Fig. 3i) lack novelty compared to: Classical theories of filler-matrix stress transfer (e.g., shear-lag model). Recent studies on mineralized collagen analogs (Ping et al., *Science* 2022, 376:188).

Response: The PVA-NC/FT/S hydrogel features PVA-NC hybrid agglomerates embedded within a continuous PVA matrix, with PVA chains spatially interconnected across both the agglomerates and matrix domains. This distinct structure enables extended load-transfer lengths and substantial energy dissipation, resulting in exceptional mechanical performance.

Such reinforcement mechanisms fundamentally differ from those of conventional inorganic-platelet-reinforced composites that generally follow classical theories such as the shear-lag model. In typical composites, applied stress is primarily transferred from the polymer matrix to individual platelets, and the composite materials are strengthened and stiffened. Under this mechanism, an increase in energy dissipation area (also called process/bridging zone) is limited, and consequently, the enhancement in stretchability and toughness is also limited. In many cases, the enhancement in strength and stiffness often come at the expense of stretchability and toughness.

In contrast, the spatial interconnectivity of polymer chains across the polymer matrix and polymer-particle agglomerates in the PVA-NC/FT/S hydrogel allows deformation of both agglomerates and the surrounding polymer network, accompanying substantial energy dissipation over large load-transfer lengths, during loading. In other words, this distinct internal structure extends the load transfer lengths and broadens the energy dissipation area. This design and reinforcement mechanisms resemble partially mineralized tissues, where stiff mineral-collagen complexes are intertwined within a flexible collagen matrix, rather than simply embedding stiff inorganic mineral/particle inside the polymer matrix as predicted by classical models.

Accordingly, we have revised the manuscript to clarify these points as follows.

(Revised manuscript, Page 4) Additionally, partially mineralized collagen-based tissues (e.g., enthesis) exhibit greater load-bearing capacity and higher stiffness than unmineralized ones, in which stiff mineralized collagens (i.e., mineral-collagen complexes) are intertwined within flexible collagen matrices²⁸⁻³⁰. Inspired by these biological structures and mechanisms, which confer high mechanical properties, we designed ligament-like strong, stiff, and tough hydrogels comprising hydrogel fibers enhanced by a distinct structure of polymer-nanoparticle hybrid agglomerates within continuous polymer matrices.

(Revised manuscript, Pages 8-9) Based on these results, including FTIR, DSC, and SEM analyses, the mechanical reinforcement and fracture mechanisms of the PVA-NC/FT/S hydrogels are proposed as follows (Fig. 3i). Upon tensile loading, the applied stress is effectively transferred from the flexible PVA matrix to the rigid PVA-NC agglomerates through strong cohesive interactions, including hydrogen bonding and spatial confinement between PVA chains and NC particles^{36-38,70}. Because the PVA-NC hybrid agglomerates are intertwined within the continuous PVA matrix and the PVA chains are spatially connected across the agglomerates and matrices, stretching of PVA chains under the loading accompanies the deformation of agglomerates. Over these extended load-transfer lengths, substantial energy

dissipation occurs^{29,30,36,37}, resulting in the hydrogel's high fracture strength and strain (Fig. 3a). With further stretching, microcracks are likely to initiate within the PVA matrix and propagate along the agglomerate surfaces, delaying ultimate failure^{36,37}. This sequential fracture mechanism imparts significant tensile strength and elastic modulus while impeding failure and enhancing toughness, overcoming the conventional trade-off between strength/stiffness and toughness/stretchability.

Importantly, these reinforcement mechanisms differ from those of conventional inorganic platelet-reinforced composites. In typical composites, inorganic platelets enhance strength and stiffness primarily through load transfer described by the shear-lag model, where applied stress is predominantly transferred from the polymer matrix to individual reinforcements via interfacial shear^{67,71-73}. Under this mechanism, the energy dissipation zone (or process/bridging/damage zone) is inherently limited, resulting in restricted toughness and stretchability^{67,74-76}. By contrast, the PVA-NC/FT/S hydrogels feature polymer-particle hybrid agglomerates embedded within a continuous polymer matrix, where polymer chains are spatially interconnected across the agglomerates and matrices throughout the entire hydrogel. This distinct structure enables effective load transfer across long distances, which leads to exceptional toughness and stretchability. If NC were only individually dispersed or poorly integrated in the matrix, load transfer would be confined to short distances, substantially diminishing mechanical reinforcement and, in particular, toughness. This structural design resembles that of partially mineralized tissues—an intermediate between unmineralized tissue (e.g., tendon) and fully mineralized tissue (e.g., bone)—which are known to achieve high strength, stiffness, and toughness beyond predictions of classical theories like the shear-lag model⁷⁷⁻⁷⁹. Therefore, the distinct internal structure enabled our hydrogel to simultaneously achieve high strength, stiffness, toughness, and stretchability, breaking the traditional trade-off observed in polymer composites.

(Added references)

71. Das, P. et al. Nacre-mimetics with synthetic nanoclays up to ultrahigh aspect ratios. *Nat. Commun.* **6**, 5967 (2015).
72. Gao, H. Application of Fracture Mechanics Concepts to Hierarchical Biomechanics of Bone and Bone-like Materials. *Inter. J. Fract.* **138**, 101-137 (2006).
73. Ji, D., Nguyen, T. L. & Kim, J. Bioinspired Structural Composite Hydrogels with a Combination of High Strength, Stiffness, and Toughness. *Adv. Funct. Mater.* **31**, 2101095 (2021).
74. Ducrot, E., Chen, Y., Bulters, M., Sijbesma, R. P. & Creton, C. Toughening Elastomers with Sacrificial Bonds and Watching Them Break. *Science* **344**, 186-189 (2014).
75. Tang, J., Chen, X., Liu, F., Suo, Z. & Tang, J. Why are soft collagenous tissues so tough? *Sci. Adv.* **11**. eadw0808 (2025).
76. Ritchie, R. O. The conflicts between strength and toughness. *Nat. Mater.* **10**, 817-822 (2011).
77. Golman, M. et al. Toughening mechanisms for the attachment of architected materials: The mechanics of the tendon enthesis. *Sci. Adv.* **7**. eabi5584 (2021).
78. Alcântara, A. et al. Molecular-Scale Interactions at Mineralized Collagen Interfaces Prevent Network Percolation, Preserving Compliance. *ACS. Nano* **19**, 31350-31362 (2025).
79. Seknazi, E. & Pokroy, B. Residual Strain and Stress in Biocrystals. *Adv. Mater.* **30**, e1707263 (2018).

#5. Statistical Rigor: Mechanical property data (e.g., ± 8 MPa for strength) should specify sample size ($n \geq 5$ recommended).

Response: Thank you for the important comment. We added sample size for all mechanical testing data and revised the manuscript.

(Revised manuscript, Experimental section) Hydrogel specimens ($n \geq 5$) were evaluated in a fiber shape (~0.4 mm diameter and ~50 mm length, and ~10 mm gauge length at the machine).

(Revised manuscript, Figure 2 legend) (h) Tensile strength and elastic modulus of the PVA-NC/FT/S hydrogel fibers ($n \geq 5$, error bars correspond to standard deviations) processed by single FT cycle and 1.5 M Na₃Citrate salt solution.

(Revised manuscript, Figure 3 legend) (a) Stress-strain curve, (b) tensile strengths and elastic moduli, and (c) toughness of PVA-NC/FT and PVA-NC/FT/S hydrogel fibers at varying concentrations of Na₃Citrate ($n \geq 5$, error bars correspond to standard deviations).

#6. Biological Relevance: Ligament-matching mechanics are highlighted, but cytocompatibility, degradation, or in vivo safety data are absent.

Response: The primary goal of this study is to develop hydrogel fibers with superior mechanical properties to those of previous hydrogels and eventually to achieve the ligament-level strength (a few tens of MPa) and stiffness (a few tens to a hundred MPa) without compromising toughness (a few tens to a hundred MJ m⁻³) that has never been accomplished before. In this goal, we employed PVA as a base material because PVA is a well-known biocompatible material and widely utilized for biomedical applications [R1, R2]. In the case of NC, as responded in the first comment, the NC employment was originally intended to enhance rheological and mechanical properties, but owing to its good biocompatibility, NC has been widely utilized in many biomedical applications [R3, R4]. Therefore, our hydrogels composed of PVA and NC are expected to be biocompatible.

[R1] Chen, M., Wu, B., Chen, A., Tucker, A., Jagota, A. & Yang, S. Fast, strong, and reversible adhesives with dynamic covalent bonds for potential use in wound dressing, *Proc. Natl. Acad. Sci. U.S.A.* **119**, e2203074119 (2022)

[R2] Cong, R. et al. Dimeric copper peptide incorporated hydrogel for promoting diabetic wound healing. *Nat. Commun.* **16**, 5797 (2025).

[R3] Yang, J. et al. A mechanical-assisted post-bioprinting strategy for challenging bone defects repair. *Nat. Commun.* **15**, 3565 (2024).

[R4] Kim, Y. et al. Bisphosphonate nanoclay edge-site interactions facilitate hydrogel self-assembly and sustained growth factor localization. *Nat. Commun.* **11**, 1365 (2020).

Although we could not perform additional biological experiments in this revision, we believe that our hydrogels with ligament-level mechanical properties hold promise for a wide range of applications, including soft robots, conductive gel electrolytes, and certain biomedical uses. Exploring these directions will be the subject of future work, particularly to expand the utility of the developed hydrogels in fiber-based or other 3D architectures. In pursuing biomedical applications, comprehensive evaluation of cytocompatibility and biodegradation will be essential.

(Revised manuscript, Page 11) This versatility enables the creation of hydrogels in customized shapes (Supplementary Figs. 5 and 6), making the material adaptable to a wide range of applications. For enhanced practical usability, the hydrogels can be coated or encapsulated with soft, thin elastomer layers⁸⁴. Because the salting-out effect primarily induces physical crosslinking within the PVA network, exposure to pure water leads to swelling and partial de-crosslinking, thereby weakening the mechanical properties (Supplementary Fig. 7). However, these properties can be effectively preserved when the hydrogel is protected by a hydrophobic layer (e.g., elastomer film and oil layer) (Supplementary Fig. 7). Given their ligament-level mechanical performance, these hydrogels hold significant potential for diverse applications, including soft robotics, conductive gel electrolytes, and bioengineering uses.

(Added references)

82. Yuk, H., Zhang, T., Parada, G. A., Liu, X. & Zhao, X. Skin-inspired hydrogel-elastomer hybrids with robust interfaces and functional microstructures. *Nat. Commun.* **7**, 12028 (2016).

#7. Strain Sensing: The $\Delta R/R_0$ response (Fig. 4g,i) lacks quantification of sensitivity/gauge factor.

Response: According to the Reviewer's suggestion, we quantified the sensitivity by gauge factor of the 8-fibe-braided hydrogel bundle over a 0–100% deformation (Supplementary Fig. 4).

Supplementary Fig. 4. The relative resistance variation ($\Delta R/R_0$) of the 8-fiber-braided hydrogel bundle over 0–100% tensile deformation. The gauge factor (GF), which is defined as the slope of the $\Delta R/R_0$ versus applied strain, is 0.1978.

(Revised manuscript, Page 9) In addition to their robust mechanical properties, the hydrogel bundles demonstrated the potential for a strain sensor (Figs. 4f–i, and Supplementary Fig. 4).

Reviewer #2

General Comment: This work integrates poly(vinyl alcohol) (PVA) with nanoclay (NC) and employs a combination of shear-induced alignment, freezing-thawing, and salting-out strategy

to fabricate hydrogel fibers. The hydrogel fibers exhibit excellent mechanical properties, with high strength (61 ± 8 MPa), stiffness (131 ± 15 MPa), and toughness (135 ± 11 MJ/m³). The fiber bundles mimic the structure of ligaments and show enhanced load-bearing capacity. The interaction between PVA matrix and NC is revealed by XRD, FTIR and DSC. Overall, this is a well-executed and impactful study with strong mechanical and structural results. One minor suggestion is to provide the water content and swelling ratio of the resulting fibers to understand the network structure and hydration behavior.

Response: We appreciate the Reviewer's encouraging comment. According to the Reviewer's suggestion, we investigated the water content (Supplementary Fig. 3) and swelling ratio (Supplementary Fig. 7) of the PVA-NC/FT/S hydrogel fibers, and the corresponding results were added in the revised manuscript. The added/revised texts and figures in the revised manuscript have been highlighted in yellow.

Supplementary Fig. 3. Water content of two different hydrogels. Error bars correspond to standard deviations at $n \geq 5$.

(Revised manuscript, Page 7) Further, the presence of larger crystalline domains in PVA-NC/FT/S, evidenced by a sharper PVA crystalline peak compared to that of PVA-NC/FT (Fig. 3f), supports superior mechanical properties of PVA-NC/FT/S to those of PVA-NC/FT (Figs. 2f and 2g)^{15,16,41}. This observation also correlates with the difference in water content between PVA-NC/FT/S and PVA-NC/FT (Supplementary Fig. 3). The larger crystalline domains indicate closer packing and stronger fastening of polymer chains within the hydrogel, which explains the lower water content observed in PVA-NC/FT/S than PVA-NC/FT.

(Revised manuscript, Experimental section) The water content of the hydrogels is measured by the difference between their weights before and after drying. The weight of the hydrogel before drying (m_0) was measured once the superficial water was wiped, and the weight of the hydrogel after drying (m_d) was measured once dehydrated in an oven at 70 °C for 72 h. The water content was calculated as $[(m_0 - m_d)/m_0] \times 100\%$.

Supplementary Fig. 7. (a) Stress-strain curves of PVA-NC/FT/S hydrogel after immersed in mineral oil and in pure water, respectively. (b) Swelling ratio of the hydrogel in pure water. (c) Tensile strength of two different hydrogels. Error bars correspond to standard deviations at $n \geq 5$.

(Revised manuscript, Page 11) Because the salting-out effect primarily induces physical crosslinking within the PVA network, exposure to pure water leads to swelling and partial de-crosslinking, thereby weakening the mechanical properties (Supplementary Fig. 7). However, these properties can be effectively preserved when the hydrogel is protected by a hydrophobic layer (e.g., elastomer film and oil layer) (Supplementary Fig. 7). Given their ligament-level mechanical performance, these hydrogels hold significant potential for diverse applications, including soft robotics, conductive gel electrolytes, and bioengineering uses.

(Revised manuscript, Experimental section) The swelling ratio of the hydrogel was measured by the difference between their weights before and after reaching the equilibrium state in water. From the weight before swelling (m_0) and after swelling (m_s), the swelling ratio was calculated as $[(m_s - m_0)/m_0] \times 100\%$.

Reviewer #3

Specific Comments

#1. The format of references is not uniform.

Response: Thanks for the Reviewer's comment. We have carefully revised the format of references.

#2. In Fig. 4a, I cannot understand why the two samples varied so much in the size and which is the braided bundle.

Response: We apologize for the confusion caused by the previous Fig. 4a. The images on the left and right sides in Fig. 4a are from the same braided samples but with different magnification. The scale bars on the left-side images are 1 cm, whereas the scale bars on the right-side images are 1 mm. We have revised Fig. 4a accordingly to clarify this distinction.

Fig. 4. (a) Photographs of hydrogel bundles formed with multiple (two to eight) fibers.

#3. I cannot what is the meaning of "the melting temperature became ill-defined". The authors should provide more explanation for this.

Response: In the original manuscript, "the melting temperature became ill-defined" means that the melting peak of the hydrogel was not defined precisely due to significant broadening by the high NC content. To clarify the meaning, we revised the sentence as follows.

(Revised manuscript, Page 8) At 8 wt% NC, the melting peak **was significantly broadened and not precisely defined**, reflecting significantly restricted thermal motion of PVA chains. This reduction in chain mobility was attributed to the spatial confinement of PVA chains between NC, a phenomenon intensified in the PVA-NC agglomerates (Fig. 1e)⁶⁷⁻⁶⁹.

#4. The authors should provide more details for the formation of braided hydrogel fiber bundles.

Response: According to the Reviewer's suggestion, we improved the Experimental section as follows.

(Revised manuscript, Experimental section) Preparation of artificial ligaments comprising multiple PVA-NC fibers. **2, 4, 8, 50 PVA-NC fibers obtained after the FT process were parallelly aligned at the same length of approximately 10 cm. The aligned fibers were then twisted to obtain one braided hydrogel bundle. The two ends of the braided hydrogel bundle were clipped to prevent untied and then submerged in 2.8 M Na₃Citrate solution for 24 h.**

#5. The preparation of hydrogel is too simple.

Response: In the revised manuscript, the preparation procedure for the hydrogel fibers has been described in detail in the Experimental section, including the exact weights of all chemicals used, and is further illustrated step-by-step in Figure 2a.

Fig. 2. (a) Schematic illustration depicting hydrogel fiber fabrication procedure: extrusion, freezing-thawing (FT), and salting-out (S) processes.

(Revised manuscript, Experimental section) Preparation of PVA-NC and pure PVA hydrogels. An aqueous 15 wt% PVA solution was first prepared by mixing 15 g PVA with 85 g water at 100 °C for 3 h. For PVA-NC mixtures with a 2, 4, 6, 8, or 10 wt% NC content, 0.204, 0.416, 0.625, 0.869, or 1.111 g NC was thoroughly mixed with the 10 g of 15 wt% PVA solution using a thinky mixer. The resultant mixture was loaded into a 10 mL plastic syringe, and bubbles in the syringe were removed using the centrifuge at 5,000 rpm for 10 min. The PVA-NC mixture was extruded into a fiber shape onto a flat glass plate using a 600 μm-diameter nozzle (Fig. 2a, extrusion process) unless otherwise defined. The extruded PVA-NC mixture underwent freezing (8 h)–thawing (8 h) for 1–5 cycles (Fig. 2a, freezing-thawing process for PVA-NC/FT hydrogels), and subsequently was immersed in a salt solution for the salting-out treatment for over 24 h (Fig. 2a, salting-out process for PVA-NC/FT/S hydrogels).

#6. We recommend the authors pay attention to some research work, such as Chemical Engineering Journal, 2023, 463, 142414.

Response: According to the Reviewer's recommendation, we additionally cited research articles in the Introduction part of the revised manuscript.

(Added references)

24. Deng, Y., Yang, M., Xiao, G. & Jiang, X. Preparation of strong, tough and conductive soy protein isolate/poly(vinyl alcohol)-based hydrogel via the synergy of biomineralization and salting out. *Int. J. Biol. Macromol.* **257**, 128566 (2024).
25. Ma, H. *et al.* Strong and tough conductive silk fibroin/poly(vinyl alcohol) composite hydrogel by a salting-in and salting-out synergistic effect. *Iranian. Poly. J.* **33**, 1527-1537 (2024).
26. Zhang, L., Wang, K., Weng, S. & Jiang, X. Super strong and tough anisotropic hydrogels through synergy of directional freeze-casting, metal complexation and salting out. *Chem. Eng. J.* **463** (2023).

#7. "ml min⁻¹" should be changed into "mL min⁻¹"

Response: Thank you for pointing that out. We modified such things, including mL min⁻¹, in the revised manuscript.

#8. The illustration in Fig.2a is not clearly. I cannot find the specific interaction between the PVA and NC.

Response: In Fig. 2a, we intended to explain the fabrication procedure (extrusion, freezing-thawing, and salting-out processes) for PVA-NC/FT/S hydrogel fibers; therefore, we did not include an illustration of the specific interactions, such as hydrogen bonds, between PVA and NC. Instead, in Fig. 3i, we clearly illustrate the interactions, hydrogen bonding and spatial confinement, between PVA and NC to describe the role of NC as mechanical reinforcement. The interactions between these components were verified through FTIR and XRD analyses, and the consequential mechanical reinforcement and fracture mechanisms were described in detail.

Fig. 3. (i) Schematic illustration of mechanical reinforcement and fracture mechanisms of the PVA-NC/FT/S hydrogel.

(Revised manuscript, Pages 8–9) Based on these results, including FTIR, DSC, and SEM analyses, the mechanical reinforcement and fracture mechanisms of the PVA-NC/FT/S hydrogels are proposed as follows (Fig. 3i). Upon tensile loading, the applied stress is effectively transferred from the flexible PVA matrix to the rigid PVA-NC agglomerates through strong cohesive interactions, including hydrogen bonding and spatial confinement between PVA chains and NC particles^{36-38,70}. Because the PVA-NC hybrid agglomerates are intertwined within the continuous PVA matrix and the PVA chains are spatially connected across the agglomerates and matrices, stretching of PVA chains under the loading accompanies the deformation of agglomerates. Over these extended load-transfer lengths, substantial energy dissipation occurs^{29,30,36,37}, resulting in the hydrogel's high fracture strength and strain (Fig. 3a). With further stretching, microcracks are likely to initiate within the PVA matrix and propagate along the agglomerate surfaces, delaying ultimate failure^{36,37}. This sequential fracture mechanism imparts significant tensile strength and elastic modulus while impeding failure and enhancing toughness, overcoming the conventional trade-off between strength/stiffness and toughness/stretchability.

Importantly, these reinforcement mechanisms differ from those of conventional inorganic platelet-reinforced composites. In typical composites, inorganic platelets enhance strength and stiffness primarily through load transfer described by the shear-lag model, where applied stress is predominantly transferred from the polymer matrix to individual reinforcements via interfacial shear^{67,71-73}. Under this mechanism, the energy dissipation zone (or process/bridging/damage zone) is inherently limited, resulting in restricted toughness and stretchability^{67,74-76}. By contrast, the PVA-NC/FT/S hydrogels feature polymer-particle hybrid agglomerates embedded within a continuous polymer matrix, where polymer chains are

spatially interconnected across the agglomerates and matrices throughout the entire hydrogel. This distinct structure enables effective load transfer across long distances, which leads to exceptional toughness and stretchability. If NC were only individually dispersed or poorly integrated in the matrix, load transfer would be confined to short distances, substantially diminishing mechanical reinforcement and, in particular, toughness. This structural design resembles that of partially mineralized tissues—an intermediate between unmineralized tissue (e.g., tendon) and fully mineralized tissue (e.g., bone)—which are known to achieve high strength, stiffness, and toughness beyond predictions of classical theories like the shear-lag model⁷⁷⁻⁷⁹. Therefore, the distinct internal structure enabled our hydrogel to simultaneously achieve high strength, stiffness, toughness, and stretchability, breaking the traditional trade-off observed in polymer composites.

(Added references)

71. Das, P. et al. Nacre-mimetics with synthetic nanoclays up to ultrahigh aspect ratios. *Nat. Commun.* **6**, 5967 (2015).
72. Gao, H. Application of Fracture Mechanics Concepts to Hierarchical Biomechanics of Bone and Bone-like Materials. *Inter. J. Fract.* **138**, 101-137 (2006).
73. Ji, D., Nguyen, T. L. & Kim, J. Bioinspired Structural Composite Hydrogels with a Combination of High Strength, Stiffness, and Toughness. *Adv. Funct. Mater.* **31**, 2101095 (2021).
74. Ducrot, E., Chen, Y., Bulters, M., Sijbesma, R. P. & Creton, C. Toughening Elastomers with Sacrificial Bonds and Watching Them Break. *Science* **344**, 186-189 (2014).
75. Tang, J., Chen, X., Liu, F., Suo, Z. & Tang, J. Why are soft collagenous tissues so tough? *Sci. Adv.* **11**, eadw0808 (2025).
76. Ritchie, R. O. The conflicts between strength and toughness. *Nat. Mater.* **10**, 817-822 (2011).
77. Golman, M. et al. Toughening mechanisms for the attachment of architected materials: The mechanics of the tendon enthesis. *Sci. Adv.* **7**, eabi5584 (2021).
78. Alcântara, A. et al. Molecular-Scale Interactions at Mineralized Collagen Interfaces Prevent Network Percolation, Preserving Compliance. *ACS. Nano* **19**, 31350-31362 (2025).
79. Seknazi, E. & Pokroy, B. Residual Strain and Stress in Biocrystals. *Adv. Mater.* **30**, e1707263 (2018).

#9. Other characterizations such as stability of hydrogel fibers at various liquid phase environments should be performed.

Response: Thanks for the Reviewer's suggestion. We additionally performed the mechanical testing for the hydrogel fibers that were fully submerged and equilibrated in pure water and mineral oil, respectively (Supplementary Fig. 7). The resultant mechanical testing data and the corresponding descriptions have been added in the revised manuscript, as follows.

Supplementary Fig. 7. (a) Stress-strain curves of PVA-NC/FT/S hydrogel after immersed in mineral oil and in pure water, respectively. (b) Swelling ratio of the hydrogel in pure water. (c) Tensile strength of two different hydrogels. Error bars correspond to standard deviations at $n \geq 5$.

(Revised manuscript, Page 11) Because the salting-out effect primarily induces physical crosslinking within the PVA network, exposure to pure water leads to swelling and partial de-crosslinking, thereby weakening the mechanical properties (Supplementary Fig. 7). However, these properties can be effectively preserved when the hydrogel is protected by a hydrophobic layer (e.g., elastomer film and oil layer) (Supplementary Fig. 7).

(Revised manuscript, Experimental section) The swelling ratio of the hydrogel was measured by the difference between their weights before and after reaching the equilibrium state in water. From the weight before swelling (m_0) and after swelling (m_s), the swelling ratio was calculated as $[(m_s - m_0)/m_0] \times 100\%$.

#10. Some characterizations for Nanoclay should be provided.

Response: According to the Reviewer's suggestion, we took a SEM image of nanoclay (NC) powder (Supplementary Fig. 8). As received dry powder state, NC is a large lump. Once the NC was mixed with PVA solution and the PVA-NC mixture went through the FT and S processes, the resulting PVA-NC/FT/S hydrogels displayed PVA-NC agglomerates (Fig. 1e).

Supplementary Fig. 8. SEM image of as-received dry NC powder.

(Revised manuscript, Experimental section) As-received dry NC powder was a large lump (Supplementary Fig. 8).

(Revised manuscript, Experimental section) The morphology of hydrogels and NC powder were characterized by the scanning electron microscope (SEM, FEI Quanta FEG 250).